# Inducible transposon mutagenesis identifies bacterial fitness determinants during infection in mice

**David W. Basta** [1,5] ✉, **Ian W. Campbell** [2,3,5], **Emily J. Sullivan** [2,3], **Julia A. Hotinger**[2,3], **Karthik Hullahalli** [2,3], **Mehek Garg**[2,3] & **Matthew K. Waldor** [2,3,4] ✉

Transposon insertion sequencing (Tn-seq) is a powerful method for genome-scale forward genetics in bacteria. However, inefficient transposon delivery or stochastic loss of mutants due to population bottlenecks can limit its effectiveness. Here we have developed 'InducTn-seq', where an arabinose-inducible Tn5 transposase enables temporal control of mini-Tn5 transposition. InducTn-seq generated up to 1.2 million transposon mutants from a single colony of enterotoxigenic *Escherichia coli*, *Salmonella typhimurium*, *Shigella flexneri* and *Citrobacter rodentium*. This mutant diversity enabled more sensitive detection of subtle fitness defects and measurement of quantitative fitness effects for essential and non-essential genes. Applying InducTn-seq to *C. rodentium* in a mouse model of infectious colitis bypassed a highly restrictive host bottleneck, generating a diverse population of $>5 \times 10^5$ unique transposon mutants compared to $10–10^2$ recovered by traditional Tn-seq. This in vivo screen revealed that the *C. rodentium* type I-E CRISPR system is required to suppress a toxin otherwise activated during gut colonization. Our findings highlight the potential of InducTn-seq for genome-scale forward genetic screens in bacteria.

Transposon insertion sequencing (Tn-seq) has become the most widely used method for conducting genome-scale forward genetic screens in bacteria[1–7]. It begins with mutagenesis of a bacterial strain with a randomly integrating transposon, followed by quantification of transposon insertion sites across the genome using high-throughput sequencing. A relative paucity of transposon insertions within a locus indicates its importance for bacterial fitness in a specific condition. Tn-seq has been applied in various contexts, including the identification of genes required for bacterial fitness within animal tissues[6–8].

Tn-seq requires the generation of a diverse transposon mutant population, typically containing ~$10^5$ unique mutants. Achieving sufficient diversity is often constrained by the efficiency of transposon

delivery, requiring labour-intensive library generation for organisms with reduced abilities to take up exogenous DNA. Transposon delivery is usually accompanied by the loss of the transposase during enrichment of the mutant population, thereby halting further mutagenesis. Consequently, the diversity of the mutant population is not temporally controlled and can be reduced by population bottlenecks, which eliminate mutants due to random chance rather than selection. This limitation curtails the use of Tn-seq to study pathogenesis in most animal models, as fewer than $10^3$ bacterial cells initiate infection in the majority of cases, imposing too great a bottleneck upon the mutant pool[9–32].

We hypothesized that temporal control of transposition could overcome limitations to traditional Tn-seq, including inefficient

[1]Department of Pathology, Brigham and Women's Hospital, Harvard Medical School, Boston, MA, USA. [2]Division of Infectious Diseases, Brigham and Women's Hospital, Boston, MA, USA. [3]Department of Microbiology, Harvard Medical School, Boston, MA, USA. [4]Howard Hughes Medical Institute, Boston, MA, USA. [5]These authors contributed equally: David W. Basta, Ian W. Campbell. ✉e-mail: dbasta@bwh.harvard.edu; mwaldor@bwh.harvard.edu

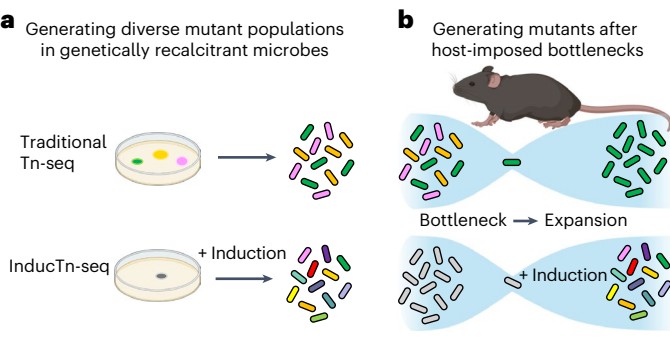

**Fig. 1 | InducTn-seq enables genome-scale forward genetics through inducible mutagenesis. a**, Traditional Tn-seq is often limited by the inefficient delivery of a transposon donor, which impedes the generation of a diverse mutant population. By contrast, InducTn-seq allows for the expansion of a single initial transconjugant (depicted as a grey colony) to generate a diverse mutant population. **b**, During animal colonization experiments, an initial host bottleneck leads to the random elimination of mutant cells, thereby reducing the diversity of the mutant population. With InducTn-seq, new transposon mutants can be generated within the animal after bypassing the initial bottleneck. Parts of this figure were created in BioRender.com.

transposon delivery or loss of mutant diversity due to host bottlenecks (Fig. 1). Inspired by work using inducible transposase expression[33–38], we have developed a streamlined approach, called 'InducTn-seq' (pronounced induction-seek), for generating maximally diverse and temporally controllable mutant populations. InducTn-seq combines inducible mutagenesis with Tn-seq, offering the following advantages over traditional Tn-seq: ease of generating highly diverse mutant populations; sensitive detection of mutants with subtle fitness defects; the ability to perform quantitative analysis of mutant fitness for genes previously characterized as essential; and the ability to circumvent host-imposed bottlenecks. We applied InducTn-seq to *Citrobacter rodentium* during infection of mice and quantified the impact of thousands of genes on bacterial fitness during experimental colitis. This in vivo screen revealed the presence of a cryptic toxin encoded within the type I-E CRISPR locus of *C. rodentium*, which is activated if the CRISPR-associated targeting complex Cascade is compromised, ensuring maintenance of the CRISPR defence system during infection.

## Results

### Design of an inducible mutagenesis system

To overcome the limitations of traditional Tn-seq, we engineered a mobilizable plasmid that introduces a randomly integrating mini-Tn5 transposon at the *att*Tn7 site in the bacterial genome (Fig. 2a, pTn donor). The mini-Tn5 transposon consists of a kanamycin-selectable marker flanked by hyperactive Tn5 mosaic ends[39] and is positioned next to its corresponding Tn5 transposase gene, which is regulated by the arabinose-responsive PBAD promoter[39,40]. This Tn5 transposition complex, composed of the mini-Tn5 transposon and transposase sequences, is nested within the left and right ends of an *att*Tn7 site-specific mini-Tn7 transposon[41].

Co-introduction of the pTn donor and a Tn7 helper plasmid (which encodes the proteins necessary for mini-Tn7 integration[42]) into a recipient strain results in integration of the Tn5 transposition complex at the *att*Tn7 site (Fig. 2a and Extended Data Fig. 1). Subsequently, culturing *att*Tn7 site-specific integrants in the presence of arabinose triggers random mini-Tn5 transposition (Fig. 2a). We note that Tn5 functions in a 'copy–paste' manner in our experiments, creating multiple copies of the transposon within a single cell's genome, consistent with previous literature[43,44].

We incorporated a Cre recombinase-based indicator of the population-level transposition frequency into the design of our system. This was achieved by flanking the Tn5 transposition complex with lox sequences, which separate an upstream constitutive promoter from a downstream gentamicin-selectable marker preceded by a transcriptional terminator (Fig. 2a and Extended Data Fig. 2). We introduced our mutagenesis system into the K-12 *Escherichia coli* strain MG1655 through conjugation, and selected for transposon integrants using kanamycin in the presence or absence of arabinose. The Cre-based indicator revealed that mini-Tn5 transposition was ~43-fold higher in the presence of arabinose, with ~28% of induced cells becoming mini-Tn5 mutants (Fig. 2b).

### High-density insertional mutagenesis with InducTn-seq

To assess the diversity of mini-Tn5 insertions following induction, we sequenced the mutant population arising from ~$10^3$ colony-forming units (c.f.u.) collected from plates with or without arabinose. Without induction, ~$1.3 \times 10^4$ unique insertions were detected, suggesting low-level transposition in the absence of inducer (that is, leakiness of the PBAD promoter). However, following induction with arabinose, ~$3.5 \times 10^5$ unique insertions were detected (Fig. 2c). These findings corroborate the Cre-based readout and provide additional evidence that arabinose induces mutagenesis. Interestingly, when cells are capable of ongoing mutagenesis during colony outgrowth, the number of unique insertions in the population exceeds the number of colonies, indicating that each colony comprises a mosaic of unique mutants.

The number of detectable unique insertions was constrained by the quantity of template DNA used in the preparation of the sequencing library (Methods). Increasing the amount of template DNA increased the number of unique insertions identified (Extended Data Fig. 3). When sampling more DNA with our inducible mutagenesis system, a diverse transposon mutant population containing over a million unique insertions could be generated from a small patch of cells scraped from a single Petri dish.

Inducible mutagenesis introduces the potential for multiple mini-Tn5 insertions within a single cell. To quantify this possibility, we performed whole-genome sequencing of ten colonies, each originating from a cell that had undergone at least one mini-Tn5 transposition event (Methods). This analysis revealed that the majority of mutant cells in the induced population contained a single mini-Tn5 insertion after overnight culture, with the likelihood of having additional insertions within the same cell decreasing exponentially, consistent with a Poisson distribution (Fig. 2d and also see 'Discussion').

### Insertions in essential genes

In contrast to traditional Tn-seq, gene-level analysis of transposon insertions revealed that virtually all genes were heavily mutagenized in the induced population, with minimal difference in insertion frequency between canonically essential and non-essential genes. For example, the genes *obgE*, *rpmA*, *rplU*, *ispB* and *murA*, which are canonically essential in *E. coli*[45–48], exhibited a similar number of insertions in the induced population as the neighbouring non-essential genes (Fig. 3a). This suggests that the rate of mutagenesis during induction surpasses the rate of selection against transposon insertions in essential genes.

We hypothesized that further culturing of the induced population in the absence of arabinose would lead to the selective depletion of transposon insertions in essential genes. To test this hypothesis, we grew the induced population under non-inducing conditions. The magnitude of depletion increased with successive generations before reaching a plateau (Fig. 3a,b and Supplementary Table 2). Based on these findings, we refer to the induced population as ON and further outgrowth in the absence of induction as OFF. The transition from ON to OFF allows us to observe the effects of selection on the mutant population. We note that ongoing low-level transposition appears to set a lower bound on the magnitude of insertion depletion (Fig. 3b), resulting in an equilibrium between selection and background transposition.

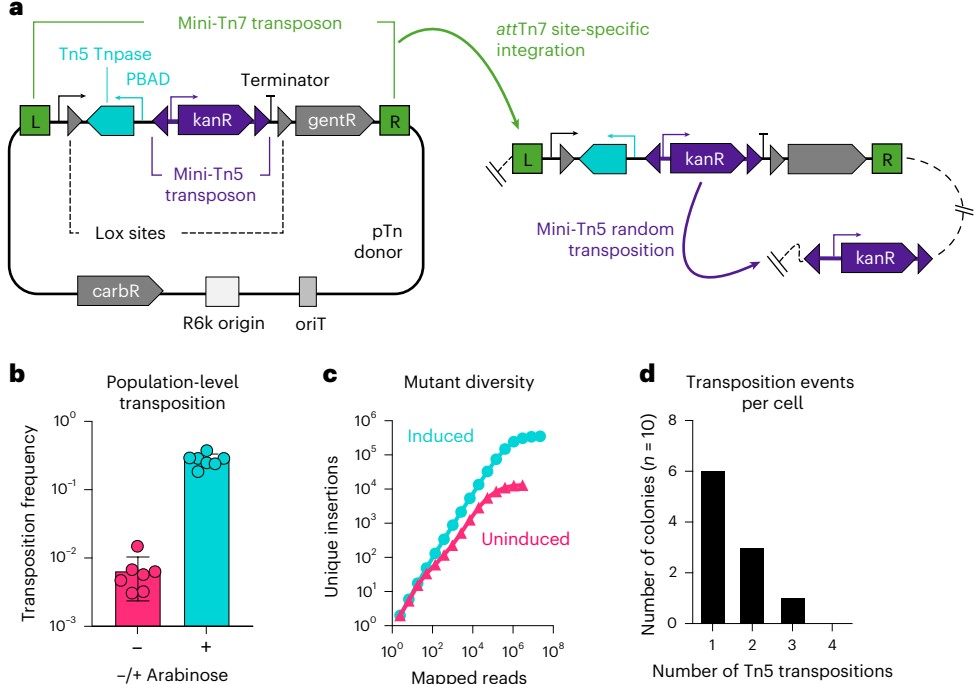

**Fig. 2 | Design of an inducible mutagenesis system. a**, Diagram of the inducible transposon mutagenesis plasmid, pTn donor. The plasmid backbone contains a carbenicillin-selectable marker (carbR), a conditional origin of replication (R6k) and an RP4 origin of transfer (oriT). The mini-Tn7 transposon (green brackets) contains a Tn5 transposase regulated by the arabinose-responsive PBAD promoter (cyan), a mini-Tn5 transposon with a kanamycin-selectable marker (kanR, purple), and a transcriptionally silenced gentamicin-selectable marker (gentR, grey). The Tn5 transposase and transposon form the Tn5 transposition complex, which is flanked by Cre-recognized lox sequences. Cre excision of the complex enables the measurement of the population-level transposition frequency (Extended Data Fig. 2 provides details). Following integration of the mini-Tn7 transposon at the *att*Tn7 site (green arrow), arabinose-mediated induction of the Tn5 transposase results in random mini-Tn5 transposition out of the *att*Tn7 site (purple arrow). **b**, The frequency of mini-Tn5 transposition out of the *att*Tn7 site after growth in the presence or absence of arabinose, expressed as the ratio of kanR + gentR c.f.u. to gentR c.f.u. (see Extended Data Fig. 2 for details). Columns represent means, error bars represent s.d., and points represent biological replicates (*n* = 7 for each condition). **c**, The number of unique mini-Tn5 insertion sites in a population of ~$10^3$ *E. coli* MG1655 colonies after growth with or without arabinose (induced or uninduced, respectively). One hundred nanograms of template DNA was used for amplification of each sequencing library. **d**, Histogram displaying the number of mini-Tn5 transposons inserted into the genome of ten colonies that underwent at least one Tn5 transposition event after arabinose induction.

## A sensitive framework for mutant fitness analysis

A direct comparison of the insertion frequency within each gene between the ON and OFF populations could simplify the analysis of mutant fitness for both essential and non-essential genes. In traditional Tn-seq, genes are classified as either essential or non-essential based on their insertion frequency relative to the genome-wide insertion density. However, this approach is susceptible to both false-positive and false-negative essential gene calls, especially in shorter, AT-rich genes and in populations with low mutant diversity[5]. By directly comparing each gene to itself between the ON and OFF conditions, InducTn-seq controls for these confounding factors. This simplified approach to essential gene analysis is fundamentally unattainable with traditional Tn-seq outside of organisms with exceptionally high transformation and recombination efficiencies[49], because insertions in essential genes are absent within the initial population.

We assessed the accuracy of InducTn-seq using this analytic framework by comparing the insertion frequency in each gene between the ON and OFF conditions. Genes with more than twofold reduction in insertion frequency ($\log_2$(fold change) < −1) and corrected $P$ < 0.01 were classified as having a fitness defect. Using these criteria, 532 of 4,494 annotated genes in *E. coli* MG1655 showed a fitness defect after dense overnight growth on solid lysogeny broth (LB) (Fig. 3c and Supplementary Table 3).

Our analysis identified a fitness defect in all 248 genes consistently classified as 'essential' across three datasets in the closely related *E. coli* strain BW25113[46–48] (Fig. 3c and Supplementary Table 4). Two comparator datasets originated from targeted gene knockout studies (Keio and Profiling of *E. coli* Chromosome (PEC)), while the third originated from traditional Tn-seq (TraDIS)[46–48]. To gauge the performance of InducTn-seq relative to traditional Tn-seq, we directly compared our dataset to TraDIS, which represents one of the most highly saturated transposon mutant populations generated in *E. coli* before this study[46]. InducTn-seq identified 331 of 354 genes classified as essential by TraDIS, a 93.5% overlap (Extended Data Fig. 4a and Supplementary Table 5). Importantly, the 23 genes identified solely by TraDIS were also not identified as essential in either the Keio or PEC datasets. To reconcile the discrepancies between our dataset and TraDIS, we quantified the gene length, GC content and fold change of genes with fitness defects identified by the two screens. The 23 genes identified only by TraDIS were shorter (median length 204 bp) and more AT-rich (median GC content 32.7%) compared to genes identified by both methods (879 bp and 52.8%) (Extended Data Fig. 4b,c and Supplementary Table 6). These data suggest that many of the genes identified solely by TraDIS may be false positives due to the lower likelihood of transposon insertions in shorter, AT-rich genes[50–53].

The 201 genes identified by InducTn-seq, but not by TraDIS, had a median $\log_2$(fold change) of −1.86 versus −3.44 for genes identified by both methods (Extended Data Fig. 4d and Supplementary Table 6). This implies that genes identified solely by InducTn-seq may have weaker fitness defects that fall below the cutoff of being classified as 'essential' by TraDIS analysis.

These results demonstrate that InducTn-seq accurately identifies known essential genes, is robust to false positives in short, AT-rich

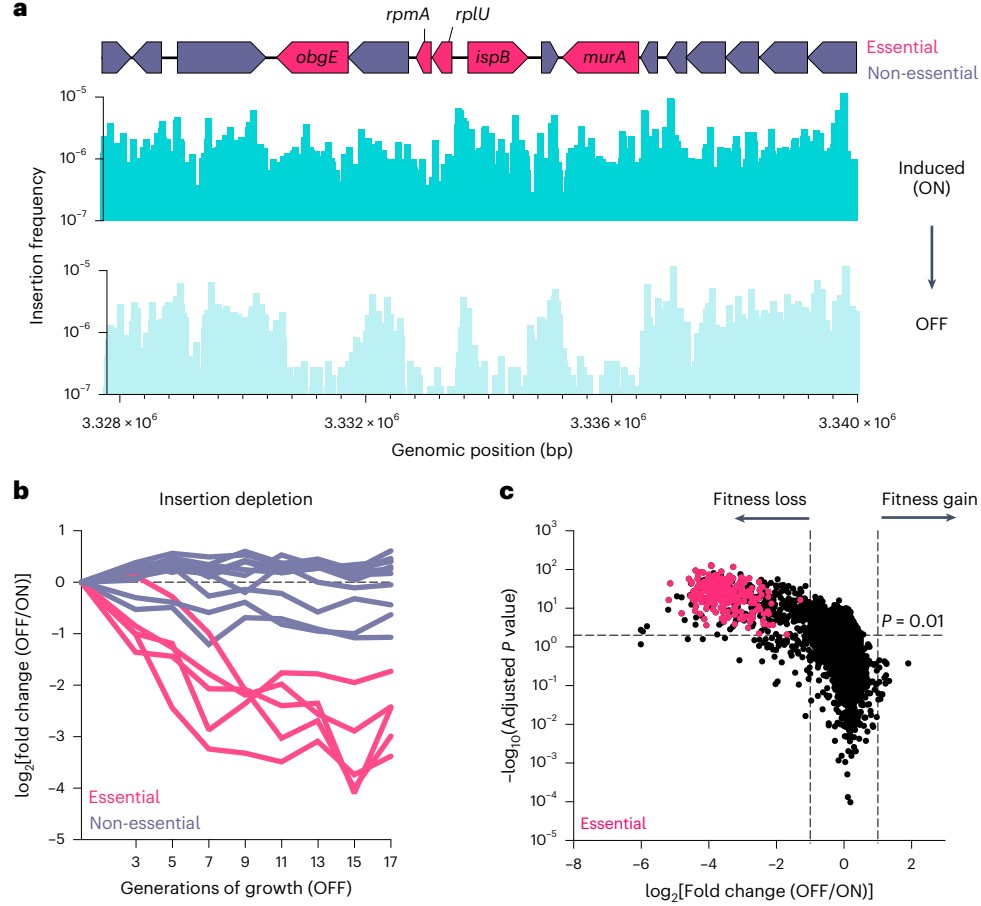

**Fig. 3 | Sensitive measurement of mutant fitness in both essential and non-essential genes. a**, Induced cells (ON) contain mini-Tn5 insertions in genes classified as essential in the closely related *E. coli* strain BW25113 (genes depicted in red, for example, *obgE*)[46–48]. **a,b**, Insertions in essential genes are selectively depleted when the population is expanded in the absence of induction (**a**), and progressively decrease with more generations of growth (**b**). In **b**, the ON population was serially diluted in LB without induction, ensuring logarithmic growth of the population over 17 generations. Individual lines in **b** correspond to the genes displayed in **a**. Supplementary Table 2 provides a complete list of all genes. **c**, Volcano plot comparing the fold change in the gene insertion frequency between OFF and ON. A significant fitness defect was defined as a two-sided Mann–Whitney *U*-test *P* value of <0.01 (adjusted for multiple comparisons with the Benjamini–Hochberg correction) and log₂(fold change) < −1 relative to the frequency of insertions in the ON population. Genes previously classified as essential are marked as red points.

genes, and can sensitively detect subtle fitness defects. Moreover, by having an ON-to-OFF comparator, InducTn-seq transforms the traditionally binary classification of 'essential' versus 'non-essential' into a quantitative measurement of fitness[54].

### High-diversity mutant populations from a single colony

We next assessed the versatility of InducTn-seq outside of K-12 *E. coli* by generating mutant populations in four enteric pathogens (*Citrobacter rodentium*, enterotoxigenic *E. coli*, *Salmonella enterica* serovar Typhimurium and *Shigella flexneri*). We introduced the pTn donor through conjugation and selected for transposon integrants. Although these pathogens are all related gammaproteobacteria, there was substantial variation in transposition frequency, spanning nearly four orders of magnitude across species (Fig. 4a). To test the power of inducible mutagenesis and mimic the bottleneck these pathogens might encounter during animal colonization, we experimentally bottlenecked the population by picking a single transconjugant colony of each strain and streaking it onto a new plate with arabinose to generate the ON population (Fig. 4b). We created transposon mutant populations in each pathogen containing 10⁵–10⁶ unique insertions (Fig. 4c). These findings show the broad applicability of InducTn-seq across species and its capacity to overcome bottlenecks that reduce the population to a single cell.

### InducTn-seq circumvents the host bottleneck

*Citrobacter rodentium* is commonly used to model human colitis because it shares an infection strategy with the human pathogens enterohaemorrhagic *E. coli* and enteropathogenic *E. coli*, but can infect and cause diarrhoea in mice without the need for antibiotic pretreatment. These pathogens all encode the conserved locus of enterocyte effacement (LEE) pathogenicity island, which enables attachment to the colonic epithelium[55,56]. A severe bottleneck impedes *C. rodentium* intestinal colonization, where only ~1–100 unique cells initiate a typical infection in mice[21,22]. Genome-scale Tn-seq libraries usually contain >10⁵ unique mutants, so traditional Tn-seq is not feasible in this model due to the random elimination of most mutants by the restrictive bottleneck. We hypothesized that inducing mutagenesis in the bacterial population after this bottleneck would enable identification of the genetic requirements for *C. rodentium* colonization. We performed InducTn-seq in C57BL/6J mice and compared it to traditional Tn-seq. For the traditional method, mice were inoculated with ~10¹⁰ cells comprising ~3 × 10⁵ unique *C. rodentium* transposon mutants. For InducTn-seq, we administered a similar number of uninduced cells and induced mutagenesis by providing mice with 5% arabinose in their drinking water from days 3 to 8. The *C. rodentium* population was tracked by enumerating faecal c.f.u. Notably, arabinose induction did not affect the abundance of faecal *C. rodentium* compared to mice not given

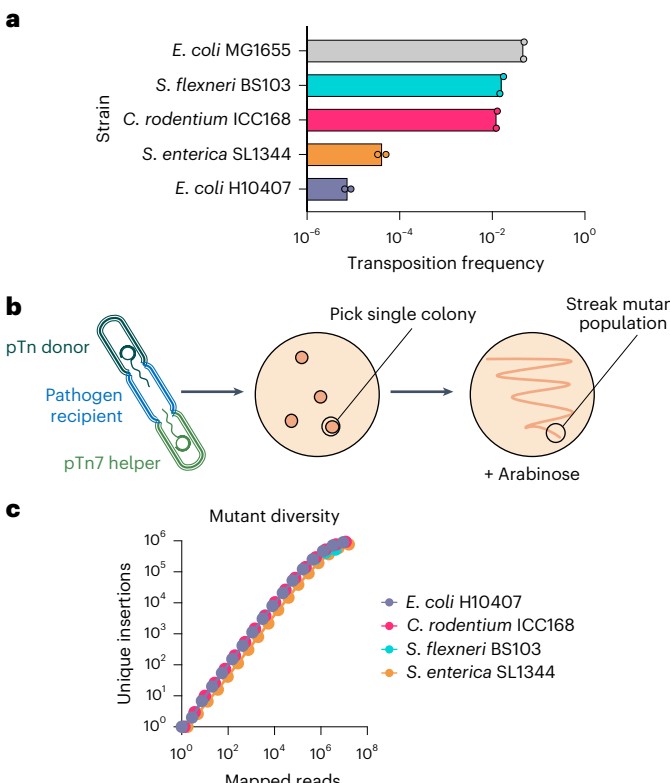

**Fig. 4 | InducTn-seq creates high-density mutant populations from a single bacterium. a**, The indicated enteric pathogens underwent conjugation with pTn donor and pTn7 helper. The transposition frequency is expressed as the ratio of kanR c.f.u. to total c.f.u. The columns represent means and the points represent biological replicates ($n = 2$ for each strain). **b**, A single, uninduced transconjugant colony of each strain was streaked onto a plate containing 0.2% arabinose. **c**, The number of unique mini-Tn5 insertion sites detected by sequencing of the streaked populations expanded from a single colony.

arabinose-containing water and infected with either the InducTn-seq population or the traditional Tn-seq population (Fig. 5a).

We then compared the transposition frequency between uninduced and induced populations in culture and within the faeces of infected mice. Similar to *E. coli*, transposition in *C. rodentium* during culture was primarily dependent on arabinose supplementation (Fig. 5b, compare to Fig. 2b). Within mice, background transposition frequency without arabinose supplementation was higher than in culture, probably due to the presence of arabinose in the mouse chow (Fig. 5b). However, arabinose supplementation in the drinking water further increased the transposition frequency approximately fourfold, and 42% of *C. rodentium* cells shed in the faeces contained at least one transposon insertion (Fig. 5b). These results indicate that transposon mutants primarily arise in an arabinose-dependent manner in vivo and that adding arabinose to the drinking water results in a high frequency of mutagenesis during infection.

Whole-genome sequencing of nine colonies expanded from mouse faeces, each having undergone at least one mini-Tn5 transposition event (Methods), revealed that four out of nine colonies contained a single mini-Tn5 insertion (Fig. 5c). This ratio is qualitatively similar to that observed for *E. coli* after overnight growth in culture with arabinose (Fig. 2d). We emphasize that multiple insertions within a single cell are unlikely to confound Tn-seq analysis ('Discussion').

As expected, the diversity of the traditional Tn-seq mutant population was dramatically reduced during infection, resulting in ~10–10² unique mutants in different mice 5 days post-inoculation (Fig. 5d). The identity and abundance of recovered mutants varied markedly between animals (average $R^2 = 0.24$; Fig. 5e,f). In contrast, induction

of mutagenesis in vivo resulted in a highly diverse mutant population within the infected animals, allowing recovery of >5 × 10⁵ unique mutants 8 days post-inoculation (Fig. 5d), with consistent gene-level insertion frequencies between animals (average $R^2 = 0.89$; Fig. 5e,f). The strong correlation between animal replicates provides additional evidence that the results were not confounded by multiple mutants within a single cell. In conclusion, InducTn-seq resolves the bottleneck problem that prevents traditional in vivo transposon mutant screens.

## InducTn-seq identifies *C. rodentium* colonization factors

We next compared the mutants induced in the mouse and then outgrown on solid LB (mouse) to a population induced in culture and then outgrown on solid LB (LB) to identify genes required for colonization. Mutants with a fitness defect in mice included previously identified *C. rodentium* colonization factors, including most of the genes within the LEE pathogenicity island[55–59] and the virulence gene regulator RegA[60] (Fig. 5g and Extended Data Fig. 5; a complete list of genes is provided in Supplementary Table 7).

Kyoto Encyclopedia of Genes and Genomes (KEGG) analysis of colonization factors revealed an enrichment for genes involved in amino-acid biosynthesis (Fig. 5h), consistent with previous work[32], along with many factors not previously known to contribute to colonization. For example, KEGG enrichment highlighted that the genes *cysAWUP*, encoding the sulfate/thiosulfate ABC transporter complex, are required during infection[61] (Extended Data Fig. 5). This finding underscores the importance of sulfur-scavenging during infection.

Both anaerobic and microaerobic metabolic pathways were vital during infection. Transposon insertions in the anaerobic transcriptional regulator *fnr* or the anaerobic metabolism genes *pta*, *ackA*, *ppc* and *pflB* all significantly decreased *C. rodentium* fitness in mice (Extended Data Fig. 5). These findings indicate that some *C. rodentium* cells probably occupy a strictly anaerobic intestinal niche, similar to recent observations for the enteric pathogen *S. enterica* serovar Typhimurium[62]. Additionally, we recovered fewer mutants in *arcA* and *arcB*, encoding a two-component system active in microaerobic conditions, and *cydA* and *cydB*, encoding the high-oxygen affinity terminal oxidase previously shown to be required by *C. rodentium* for microaerobic respiration in the intestine[63].

By contrast, genes encoding the low-oxygen affinity terminal oxidase CyoABCDE, along with terminal oxidases involved in the respiration of alternative electron acceptors (that is, nitrite/nitrate, fumarate or dimethyl sulfoxide (DMSO)), were not required during infection, indicating that the concentration of these substrates in the gut was either absent or too low to support *C. rodentium* respiration (Extended Data Fig. 5). Interestingly, disruption of genes encoding some of these terminal oxidases, especially components of the Cyo complex, resulted in a fitness benefit during infection. This may reflect an unrecognized fitness cost associated with the metabolic versatility afforded by encoding multiple terminal oxidases.

Disruption of genes encoding the ABC transporter complex FepBDGC, required for uptake of the iron-scavenging siderophore enterobactin[64], resulted in a significant fitness advantage during infection, but a significant fitness disadvantage during growth in LB (Fig. 5g and Extended Data Fig. 5). This suggests that enterobactin functions to scavenge iron in culture but is not required by *C. rodentium* within the intestine.

In summary, InducTn-seq sensitively detects both previously characterized and uncharacterized colonization factors, highlighting the environmental challenges and constraints experienced by *C. rodentium* during enteric infection.

## A CRISPR-repressed toxin is active during infection

Intriguingly, InducTn-seq identified all components of the type I-E CRISPR Cascade complex (CasABCDE) as gut colonization factors (Fig. 6a and Extended Data Fig. 5). Cascade is a protein complex that,

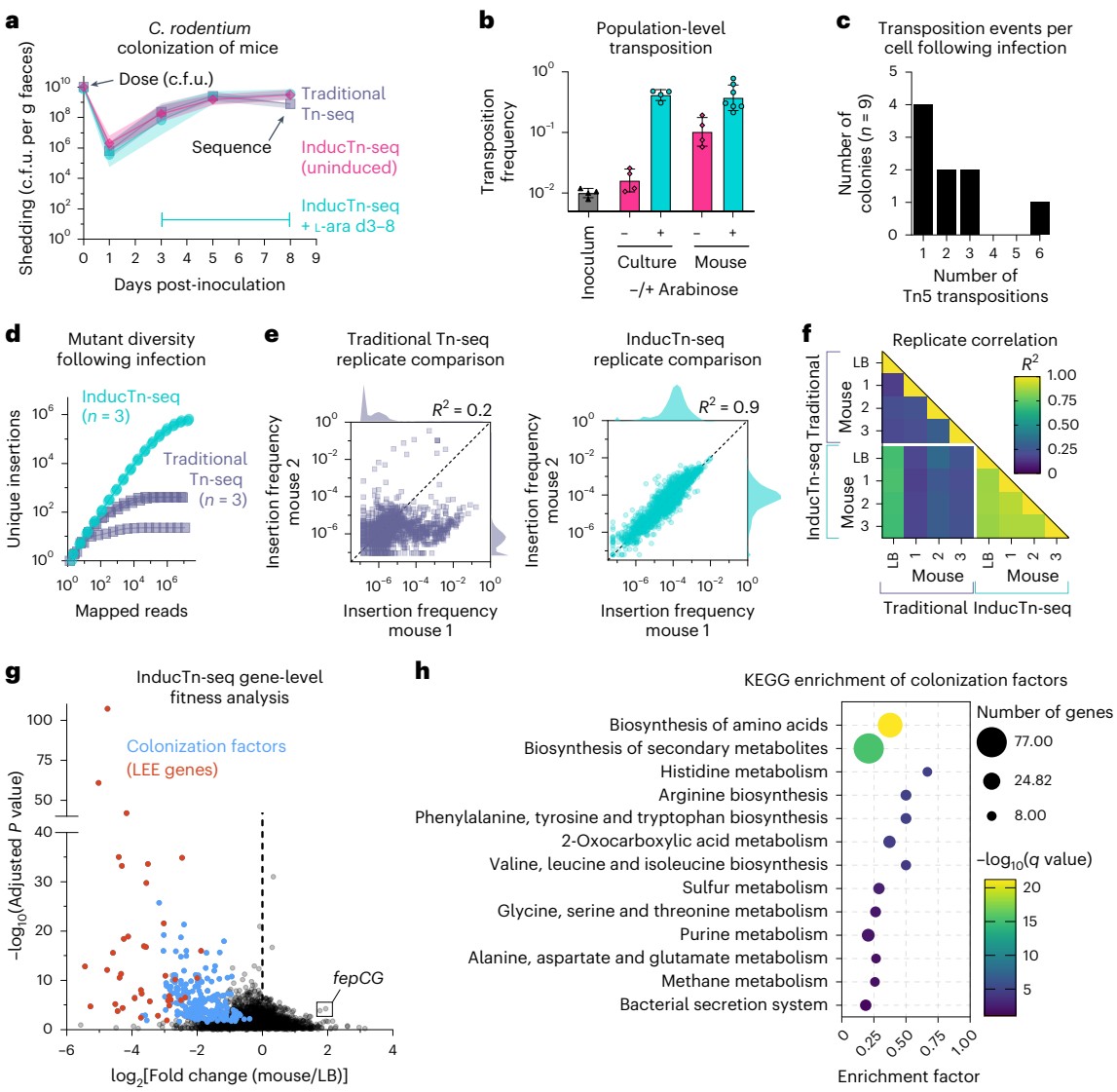

**Fig. 5 | InducTn-seq bypasses the host bottleneck to enable identification of *C. rodentium* colonization factors. a**, Female C57BL/6J mice were intragastrically inoculated with either a pool of ~3 × 10⁵ unique *C. rodentium* mini-Tn5 insertion mutants (traditional Tn-seq, *n* = 4) or uninduced Tn5 transposition complex integrants (InducTn-seq). Colonization was monitored by serial dilution and plating of faeces. For InducTn-seq, mini-Tn5 transposition was induced from days 3 to 8 by providing ad libitum access to water containing 5% arabinose (induced InducTn-seq, *n* = 20; uninduced InducTn-seq, *n* = 8). No difference in log₁₀(c.f.u.) was observed between arabinose induction and either uninduced or traditional Tn-seq by two-sided restricted maximum likelihood mixed-effects analysis with Šídák's correction for multiple comparisons (adjusted *P* > 0.01). **b**, Frequency of mini-Tn5 transposition out of the *att*Tn7 site after growth of the InducTn-seq inoculum in the presence or absence of arabinose, either in culture or in mice (*n* = 4 in inoculum, culture minus arabinose, culture plus arabinose and mouse minus arabinose; *n* = 7 in mouse plus arabinose). **c**, Histogram showing the number of mini-Tn5 transposons inserted into the genome of nine colonies that underwent at least one Tn5 transposition event following arabinose induction in the mouse. **d**, Samples from day 5 (traditional Tn-seq) or day 8 (InducTn-seq)

post-inoculation were sequenced to determine the number of unique mutants recovered from each animal (*n* = 3 mice in each group). **e**, Correlation of mutant frequency between animal replicates. Points represent genes. Insertion frequency is calculated as reads per gene normalized to total reads in the sample, and histograms on the axes display the distribution of the data. **f**, Coefficient of determination (*R²*) comparing the log₁₀-transformed insertion frequencies across replicates. **g**, Volcano plot comparing fold change in the gene insertion frequency between mouse and LB. The average log₂(fold change) and −log₁₀(*P* value) of a two-sided Mann–Whitney *U*-test adjusted for multiple comparisons with the Benjamini–Hochberg correction is shown for three biological replicates. Colonization factors are shown in blue and are defined as genes meeting a cutoff of *P* < 0.01 and log₂(fold change) < −1 in at least two out of three animal replicates (Supplementary Table 7). Genes within the LEE are shown in red. **h**, KEGG enrichment analysis of colonization factors displaying pathways with *q*-value < 0.05. Enrichment factor is defined as the fraction of genes within the pathway that are colonization factors. Data in **a** and **b** are presented as geometric mean values ± s.d.

together with a CRISPR RNA guide (crRNA), recruits the nuclease Cas3 for targeted degradation of foreign nucleic acids[65]. Although genes encoding Cascade were required for fitness in the mouse, *cas3* was not required, suggesting that the role of CRISPR during colonization is unrelated to Cas3-dependent DNA degradation (Fig. 6a). Furthermore, genes encoding the spacer acquisition proteins, Cas1 and Cas2, were

also not required for colonization. We confirmed these findings by creating a deletion of one of the Cascade genes (Δ*casD*), and of the entire CRISPR locus (Δ*casABCDE123*). Notably, Δ*casD* also lacks the first seven nucleotides of *casE*, probably resulting in loss of both CasD and CasE activity. Consistent with our InducTn-seq results, Δ*casD* was unable to colonize mice. Surprisingly, however, deletion of the entire

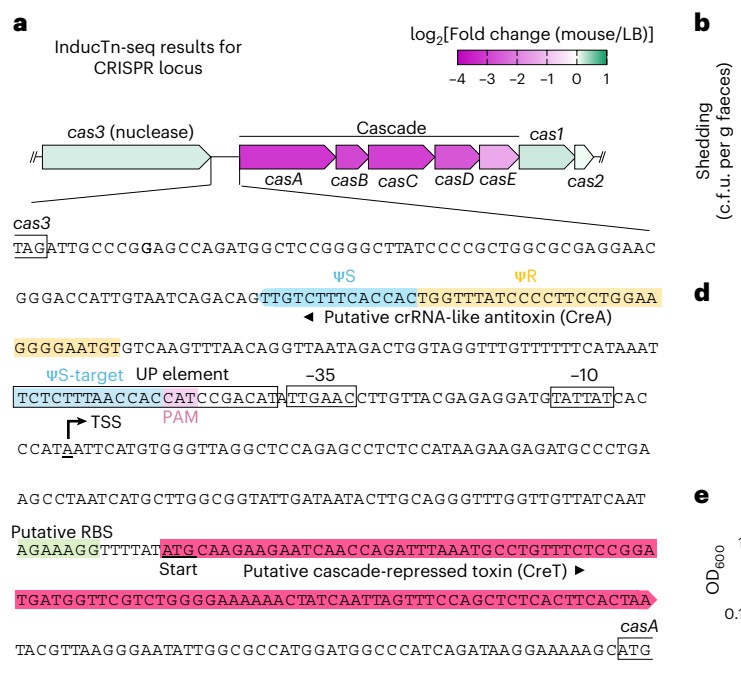

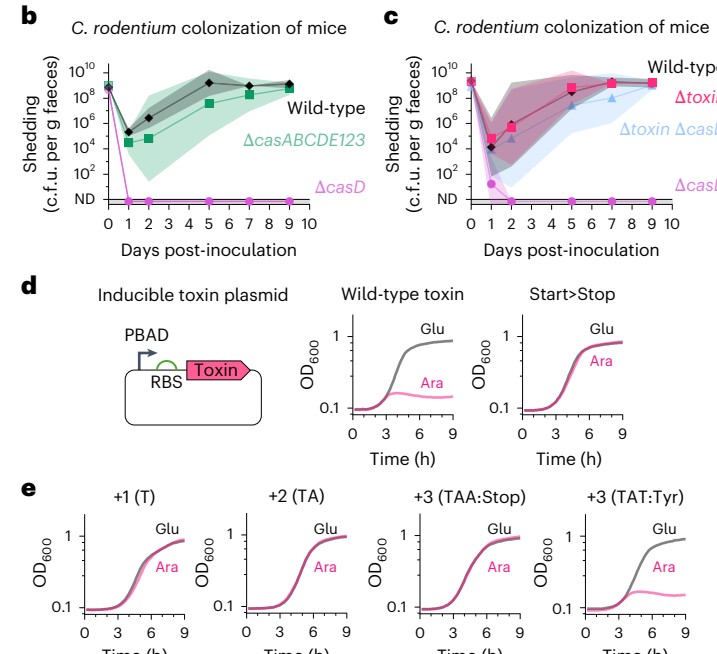

**Fig. 6 | A Cascade-repressed toxin is activated during gut colonization.**
**a**, Diagram of the *C. rodentium* type I-E CRISPR locus overlaid with a heatmap of the average log₂(fold change) in insertion frequency comparing mouse to LB. Genes encoding Cascade (*casABCDE*), but not the nuclease encoded by *cas3*, are required for fitness during enteric infection. The intergenic sequence between *cas3* and *casA* is expanded to show the putative 96-bp toxin (CreT, red). The start codon of CreT is underlined. The repeat-like (ΨR, yellow) and spacer-like (ΨS, blue) segments of the putative crRNA-like antitoxin (CreA) are highlighted. A potential target sequence (ΨS-target) with partial complementarity to ΨS overlaps with the predicted UP element and is adjacent to a canonical Cascade-recognized protospacer adjacent motif (PAM, 5′-ATG-3′). **b,c**, Bacterial shedding in female C57BL/6J mice colonized with wild-type *C. rodentium* or the

indicated deletion strains was monitored by serial dilution and plating of faeces (*n* = 4 mice per strain (**b**); *n* = 6 mice per strain (**c**)). **d,e**, The putative CreT toxin (**d**) or its indicated variants (**d**,**e**) were placed on a plasmid under the control of the arabinose-responsive PBAD promoter and transformed into *C. rodentium*. All constructs retained the predicted RBS and the upstream promoter elements were replaced by the PBAD promoter. Growth was measured for each variant in LB supplemented with 0.2% arabinose or 0.2% glucose. The average growth curve of four biological replicates derived from independently picked transformants is shown. Nucleotides (+1/+2/+3) were inserted after the putative creT start codon in **e**. OD₆₀₀, optical density at 600 nm. Data in **b** and **c** are presented as geometric mean values ± s.d.

CRISPR locus did not result in a fitness defect: Δ*casABCDE123* colonized like wild-type (Fig. 6b). As deletion of the entire CRISPR locus rescues the colonization defect of Δ*casD*, these results suggest that an epistatic interaction within the CRISPR locus causes the observed phenotype.

We hypothesized that CasD, along with the other components of Cascade, represses a toxic element within the CRISPR locus. Cascade-repressed toxins (CreT) are a recently described safeguard mechanism activated when CRISPR activity is compromised, either by phage-encoded anti-CRISPR systems or by insertional disruption of Cascade components by transposable elements[66–69]. Cascade acts in concert with a crRNA-like antitoxin (CreA) that partially base-pairs with the toxin promoter to repress toxin expression. The CreTA module is thought to serve as a 'CRISPR addiction' system that prevents the loss of Cascade activity. The first CreTA module was identified in the archaeon *Haloarcula hispanica*, where it is encoded within a 311-bp intergenic region between *cas6* (*casE*) and *cas8* (*casA*)[66].

We hypothesized that a similar CreTA module exists in *C. rodentium* and that it might be located within the 495-bp intergenic region between *cas3* and *casA* (Fig. 6a). Attempts to clone this intergenic region were unsuccessful, suggesting that the region was toxic in *E. coli*. To explore whether a cryptic toxin lies within the *cas3/casA* intergenic region, we deleted a 229-bp segment of this region in both the wild-type (Δ*toxin*) and Δ*casD* (Δ*toxin* Δ*casD*) backgrounds and repeated the mouse colonization experiments. Although deletion of this region had no impact on colonization in the wild-type background, it rescued the colonization defect of Δ*casD*, proving that this intergenic region was necessary to cause the Δ*casD* colonization defect (Fig. 6c).

*Haloarcula hispanica* CreT is a small RNA that sequesters tRNAs decoding rare arginine codons, leading to a shutdown of translation. Within the *cas3/casA* intergenic region of *C. rodentium* is a 96-bp open reading frame (ORF) downstream of a predicted strong promoter and ribosome binding site (RBS)[70–73] (Fig. 6a). BLAST analysis found that neither the nucleotide nor translated amino-acid sequence of this ORF has homology to the CreT of *H. hispanica* or to any other annotated sequence. Upstream of the ORF and on the opposite strand is a putative crRNA-like antitoxin (CreA) with partial complementarity to a nearby target sequence (ΨS-target) that overlaps with the UP element of the predicted promoter[74]. This target sequence is adjacent to a canonical type I-E protospacer adjacent motif required for Cascade binding (Fig. 6a).

To localize the toxin within the *cas3/casA* intergenic region, we placed this ORF onto a plasmid under the control of the arabinose-responsive PBAD promoter and transformed the construct directly into *C. rodentium*. Consistent with the presence of an active toxin, arabinose induction arrested the growth of cells carrying the wild-type toxin sequence in a concentration-dependent manner (Fig. 6d and Extended Data Fig. 6). Mutation of the AUG start codon of the ORF to a UAG stop codon completely restored growth in the presence of arabinose (Fig. 6d).

Unlike CreT in *H. hispanica*, the CreT in *C. rodentium* does not encode rare amino-acid codons, suggesting differences in toxin identity and function between these two distantly related organisms. To confirm that translation of the toxin ORF is necessary for its activity, we inserted 1, 2 or 3 bp after the AUG start codon and monitored the

toxin activity of these variants. Out-of-frame insertions (+1, +2) and an in-frame premature stop codon (+3 TAA) eliminated toxin activity, whereas an in-frame tyrosine codon (+3 TAT) maintained it (Fig. 6e). Thus, we conclude that *C. rodentium* encodes a Cascade-repressed toxin that is activated during gut colonization and must be translated for its activity. Intriguingly, this is the first description of a CRISPR-associated toxin–antitoxin system that is conditionally active during infection, suggesting an evolutionary imperative for *C. rodentium* to protect CRISPR activity within the intestine.

## Discussion

Despite limitations, Tn-seq is the method of choice for genome-scale forward genetics in bacteria because of its simplicity, cost-effectiveness and power for uncovering genotype–phenotype relationships. InducTn-seq retains these advantages while enabling the generation of mutants on demand. The ability to control mutagenesis temporally and with high efficiency offers several benefits over one-time transposition, including (1) the ability to generate millions of mutants from a single clone, thereby overcoming barriers to conjugation and transformation in genetically recalcitrant microbes; (2) the creation of unparalleled diversity, which enhances the sensitivity of detecting subtle phenotypes; (3) the capacity to quantify the fitness impact of canonically essential genes, moving beyond a binary classification of essentiality; (4) the ability to induce mutagenesis after host-imposed colonization bottlenecks, permitting functional genetics in vivo.

Historically, inducible mutagenesis has been avoided in functional genetic screens presumably due to the risk of multiple insertions within the same cell. Multiple insertions can obscure the causal relationship between each insertion and an observed phenotype and introduce the possibility of phenotypes arising from genetic interactions between insertions. However, pooled genetic screens like Tn-seq are inherently robust against the potential confounding effects of multiple transposon insertions within the same cell because the relationship between genotype and phenotype is predicated on the collective behaviour of tens to hundreds of unique, independently derived insertions. Leveraging multiple unique insertions eliminates the confounding effects of additional insertions in the cell on interpreting causality. Even in an extreme scenario where an entire population consists of double mutants, the phenotypic interpretation for a given gene remains unaffected. Any observed change in a gene's insertion frequency is overwhelmingly due to the insertions in the gene itself, rather than the other random insertions, which are diluted across the rest of the genome. Thus, the collective behaviour of mutants in the gene will be attributed to the gene.

Moreover, using non-parametric statistical tests in Tn-seq analysis to compare the concordant behaviour of many unique insertion sites (for example, the Mann–Whitney *U*-test[5]) limits the ability of outliers, particularly those caused by positive genetic interactions, to confound interpretation. For example, in a hypothetical condition where a genetic interaction leads to a very strong selective advantage, such as a condition where most mutants cannot grow except for the double mutant, the interaction would appear as an unreproducible 'spike' of insertions at one or a few unique sites within each interacting gene. This finding is generally classified as non-significant by non-parametric statistical tests, because the insertion frequency at this site is not concordant with the insertion frequency across the rest of the gene. Consequently, inducible mutagenesis can be used in pooled Tn-seq experiments with minimal risk of multiple insertions in the same cell confounding the results. However, if the transposon mutant population is created in a mutant rather than a wild-type background, then the role of both negative (that is, synthetic sick or lethal) and positive genetic interactions can be assessed[75,76].

While our attempt to perform a genome-scale screen in mice using a traditional *C. rodentium* transposon mutant population was unsuccessful (Fig. 5), Caballero-Flores and colleagues previously used a traditional Tn-seq approach to identify genes required for *C. rodentium* colonization[32]. Because the bottleneck to *C. rodentium* colonization is primarily dependent on the microbiota[22], the more permissive bottleneck observed in the Caballero-Flores et al. screen is probably due to variations in the composition of the murine microbiota between institutions. Treatments to neutralize gastric acid or deplete the commensal microbiota[21,22,32,77] are often used to widen the host bottleneck when implementing traditional Tn-seq in mouse models of intestinal colonization. However, these interventions also disrupt the natural dynamics between host and microbe(s), and can alter the genetic requirements for colonization. For example, the LEE is required for *C. rodentium* to colonize mice with an intact microbiota (Fig. 5g and Extended Data Fig. 5), but is dispensable in germ-free animals[78]. A major advantage of InducTn-seq is that it avoids the need to disrupt natural colonization bottlenecks.

Another advantage of InducTn-seq in animal studies is its sensitivity. By recovering >10[5] unique *C. rodentium* mutants from infected mice, we resolved subtle phenotypes and uncovered metabolic pathways that facilitate pathogenesis. We found that both anaerobic and microaerobic metabolism are required for fitness within the animal. In contrast, genes necessary for aerobic metabolism in highly oxygenated environments (*cyoABCDE*) or for scavenging iron (*fepABCDG*) were only required in culture and surprisingly imparted subtle fitness costs during infection (Fig. 5g and Extended Data Fig. 5). The ability of InducTn-seq to quantify these metabolic trade-offs provides valuable insight into the life cycle of enteric pathogens and the evolution of bacterial pathogenesis. Inducible mutagenesis will also facilitate the spatiotemporal characterization of genetic programs employed by pathogens during different stages of infection and in different tissues in animal-based studies.

Many bacterial genes remain uncharacterized because they are exclusively required by microbes in their natural environments. By expanding the range of conditions amenable to a forward genetics approach, InducTn-seq enables gene-fitness characterization in physiological contexts. For example, probing the genetic requirements for *C. rodentium* fitness within an infected animal revealed an uncharacterized toxin hidden within the CRISPR locus. CRISPR-associated toxin–antitoxin systems were previously discovered in other microbes when their constitutive activity prevented the disruption of native CRISPR systems[66]. However, the toxin–antitoxin module in *C. rodentium* appears to be conditionally active within the intestine (Fig. 6a and Extended Data Fig. 5). Enteric regulation suggests that this protective mechanism may be evolutionarily linked to heightened threats to the CRISPR locus during infection, such as from anti-CRISPR systems encoded by resident phages in the microbiota, or from endogenous transposons that become more active within the intestine. We anticipate that adapting InducTn-seq to additional natural contexts will yield similar advances in our understanding of the environment-specific functions of uncharacterized genes.

Although we used the arabinose-responsive PBAD promoter to induce transposase expression[40], we note that arabinose supplementation during colonization experiments may affect virulence, as well as microbial and host metabolism[79–81]. Fortunately, other regulatable promoter systems could be adapted for the control of mutagenesis, such as the anhydrotetracycline-responsive TetR system[82], or temperature-inducible promoters active at 37 °C that would eliminate the need to supply inducer molecules during infection of mammalian hosts[83,84]. In all cases, tight promoter regulation is important to ensure that in the absence of inducer the effects of selection are not obscured by ongoing background mutagenesis.

Although Tn7-based integration will not be applicable in many microorganisms, additional site-specific integration systems, such as phage-based integration vectors for *Mycobacterium tuberculosis* and *Listeria monocytogenes*[85,86], should be useful for chromosomal integration of an inducible transposition system. We also note that InducTn-seq is effective when expressed from a replicative plasmid

rather than being integrated into the genome, making it applicable in organisms that lack the *att*Tn7 site or other means of genomic integration. In summary, InducTn-seq should be widely adaptable to diverse microbial systems, including non-prokaryotic systems such as the human parasite *Plasmodium falciparum*, where Tn-seq has previously been applied[87].

## Methods

### Bacterial strains
The strains used in this paper are listed in Supplementary Table 1. For cloning, we used the *E. coli* strain MFD*pir*[88]. Transposon mutant populations were made in the K-12 *E. coli* strain MG1655, enterotoxigenic *E. coli* (ETEC) strain H10407, *S. enterica* serovar Typhimurium strain SL1344 and *S. flexneri* strain BS103 (a derivative of strain 2457T lacking the virulence plasmid). For animal infection experiments we used a spontaneous streptomycin-resistant isolate of *C. rodentium* strain ICC168 (ref. 22).

### Plasmid assembly
All plasmid assembly was performed using the NEBuilder HiFi DNA Assembly Master Mix (New England Biolabs, catalogue number E2621). Fragments were generated through polymerase chain reaction (PCR) amplification from plasmid or genomic DNA templates using primers acquired from Integrated DNA Technologies (IDT). The RBS located upstream of the Tn5 transposase ORF in pTn was designed using the RBS calculator v2.2 (refs. 71–73). The target translation initiation rate was set at 20,000 arbitrary units. The *cre* ORF in pCre was amplified from an arabinose-regulated Cre-expression vector[89]. Following assembly, plasmids were cloned into *E. coli* strain MFD*pir* using electroporation. Each plasmid sequence was then verified through whole-plasmid sequencing (Plasmidsaurus). All plasmid sequences were edited using the open-source software 'A plasmid Editor' (ApE)[90] and annotated using pLannotate[91]. The transposon mutagenesis plasmid pTn donor and the Cre-expressing plasmid pCre are deposited on Addgene (https://www.addgene.org/browse/article/28253166/) and are provided as supplementary GenBank files.

### Generation of induced transposon mutant populations
Induced mutant populations were generated using the donor strain MFD*pir* + pTn and the helper strain MFD*pir* + pJMP1039, which expresses the Tn7 transposase enzymes[42]. Both strains were cultured in liquid LB containing 1 mM diaminopimelic acid (DAP) and 50 µg ml⁻¹ carbenicillin, while recipient strains were cultured in liquid LB without DAP or antibiotics. The cultures were incubated overnight at 37 °C with shaking (250 r.p.m.).

The following day, 1 ml of each overnight culture was washed once with 1 ml of fresh LB, and then resuspended in 50–100 µl of LB. The cultures were mixed in a 1:1:1 ratio of MFD*pir* + pTn, MFD*pir* + pJMP1039 helper and recipient strain. A 60-µl aliquot of this mating mixture was spotted onto a 0.45-µm pore mixed cellulose ester filter disk (Millipore) placed on solid LB containing DAP and incubated at 37 °C for 2 h.

After incubation, the filter disks were removed from the plate using sterile forceps and resuspended in 1 ml of LB in a microfuge tube by vortexing. The resuspended mating mixtures were then diluted and plated on solid LB containing 50 µg ml⁻¹ kanamycin and 0.2% L-arabinose to induce Tn5 transposase expression. For the *E. coli* H10407 recipient strain, 100 µg ml⁻¹ of kanamycin was used. After overnight growth, colonies were scraped from the plates, resuspended in liquid LB containing 15% glycerol, and stored as aliquots at −80 °C.

To induce mutant populations of enteric pathogens starting from a single colony, resuspended mating mixtures were first plated on solid LB containing kanamycin without arabinose. After overnight growth, a single transconjugant colony of each strain was then streaked onto solid LB containing kanamycin and arabinose to form the induced mutant population.

### Assessment of mini-Tn5 and mini-Tn7 integration frequency
The integration frequencies of mini-Tn5 and mini-Tn7 were assessed by serial dilution and plating of the resuspended mating mixtures on solid LB with or without kanamycin to enumerate c.f.u. values[92]. The integration frequency, representing the combined total of mini-Tn5 and mini-Tn7 integration, was quantified by comparing the number of kanamycin-resistant c.f.u. to total c.f.u. To assess the frequency of mini-Tn5 random integration alone, MFD*pir* + pJMP1039 (Tn7 helper plasmid) was excluded from the mating mixture.

### Assessment of mini-Tn5 transposition frequency and transposition events per cell
The transposition frequency of mini-Tn5 in culture was assessed by plating the resuspended mating mixtures on solid LB containing 50 µg ml⁻¹ kanamycin, with or without 0.2% L-arabinose. After overnight growth, colonies were scraped from each plate and a second conjugation was performed with MFD*pir* + pCre. To measure the transposition frequency following infection, faeces were resuspended in phosphate-buffered saline (PBS), passed through a 5-µm filter, and the filtrate was conjugated with MFD*pir* + pCre. Following this second conjugation, the resuspended mixtures were serially diluted and plated on solid LB containing 20 µg ml⁻¹ gentamicin with or without 50 µg ml⁻¹ kanamycin. The transposition frequency was quantified by comparing the number of gentamicin and kanamycin-resistant c.f.u. to gentamicin-resistant c.f.u. (Extended Data Fig. 2).

Cre activity excises the Tn5 transposase, thereby preventing further mutagenesis and leaving the cell with a fixed number of mini-Tn5 insertions. To determine the number of transposition events per cell following Cre recombination, gentamicin and kanamycin-resistant colonies (that is, originating from cells that had undergone at least one transposition event) were individually streaked onto solid LB and sent to the Microbial Genome Sequencing Center (Pittsburgh, PA) for whole-genome sequencing.

### Assessment of insertion depletion in the absence of induction
The depletion of transposon insertions was assessed by performing repeated serial dilution and outgrowth of the ON population in the absence of induction (OFF). A 50-µl aliquot of a frozen glycerol stock of the *E. coli* MG1655 ON population was diluted into 5 ml of LB to achieve a starting concentration of $5 \times 10^7$ c.f.u. ml⁻¹ and incubated at 37 °C with shaking at 250 r.p.m.

After 2 h of incubation, a 2-ml aliquot of the culture was pelleted for sequencing library preparation and serial dilutions were performed to enumerate c.f.u. values. The culture was then back-diluted fourfold into fresh LB and incubated for another hour. This growth and dilution step was repeated hourly for a total of 17 generations of growth, which is approximately two generations of growth per hour. An aliquot was pelleted and c.f.u. values were enumerated at each time point.

### Sequencing library preparation
Sequencing libraries were prepared by modifying the established protocols reviewed in ref. 93. In brief, genomic DNA was extracted from pelleted aliquots of the mutant populations using the Qiagen DNeasy Blood and Tissue Kit. The extracted DNA was then sheared to a size of 400 bp using a Covaris M220 ultrasonicator. The sheared DNA was end-repaired using the NEB Quick Blunting kit, followed by the addition of Poly(A) nucleotide overhangs using Taq polymerase and dATP. Hybridized Illumina P7 adapters with complementary 'T' overhangs were then ligated to the A-tailed DNA using NEB T4 ligase.

The end of the mini-Tn5 transposon is flanked by cut sites recognized by the restriction enzymes PacI and SpeI (see annotated plasmid sequence in Supplementary Data 1). After adapter ligation, the DNA was digested with PacI and SpeI, resulting in the creation of an ~90-bp fragment containing the transposon end within the integrated vector sequence. Following digestion, the DNA was size-selected using

SPRIselect beads at a 1× ratio to remove the digested fragment. This step obviates the need to cure the integrated portion of the plasmid vector, streamlining the generation of both transposon mutant and sequencing libraries, and eliminating uninformative sequencing reads that map to the integrated vector.

Following size selection, 100–800 ng of the DNA was amplified for 20 cycles using a transposon-specific forward primer and an adapter-specific reverse primer. Each reverse primer contained a unique Illumina i7 index sequence. The PCR products underwent a second round of 1× size selection to remove any primer dimer. The concentration of the sequencing libraries was quantified using a Qubit fluorimeter and then pooled and sequenced on a Nextseq 1000. The sequencing reads are deposited in the Sequencing Read Archive (SRA) under accession number PRJNA1113708.

### Tn-seq data analysis

Sequencing reads were trimmed of the transposon end sequence and mapped to the reference genome using the Burrows–Wheeler Aligner (BWA) version 0.7.17-r1188[94]. Mapped reads were then converted into a table containing a tally of the read count at each genomic position and the corresponding annotated gene name at that position using a custom script in Python version 3.12. The counts were normalized for read depth, and the $\log_2$(fold change) in insertion frequency was calculated for each gene feature by comparing OFF to ON. To determine statistical significance, the genome-wide counts for each sample were first summed within a window size set as the inverse of the frequency of non-zero positions for the sample with fewer overall non-zero positions. Setting this window size helps control for differences in the overall diversity of transposon insertions between two samples. Following summation of the counts into equal window sizes, the non-parametric Mann–Whitney $U$ statistical test was performed, and $P$ values were adjusted for multiple testing using the Benjamini–Hochberg correction.

*Citrobacter rodentium* colonization factors identified by InducTn-seq were defined as genes meeting a cutoff of $P < 0.01$ from Mann–Whitney $U$ analysis and $\log_2$(fold change) $< −1$ in at least two out of three animal replicates. KEGG enrichment analysis was performed in R version 4.4.2 with clusterProfiler using the default settings and a $P$-value threshold of 0.1 (ref. 95).

To visualize the number of unique insertion sites per sample across different sampling depths, the table of read counts per genomic position was filtered by removing the bottom 1% of all reads. This step was performed to remove noise resulting from Illumina index sequence hopping. After this noise correction, the distribution of counts per site was subjected to multinomial resampling beginning with the maximum read count of each sample and subsampling in twofold lower increments. At each sampling depth, the number of unique insertions was calculated.

### Mouse experiments

Animal studies were conducted at the Brigham and Women's Hospital in compliance with the Guide for the Care and Use of Laboratory Animals and according to protocols reviewed and approved by the Brigham and Women's Hospital Institutional Animal Care and Use Committee (protocol number 2016N000416). Adult (9–12 weeks), female, C57BL/6J mice were purchased from Jackson Laboratory (strain number 000664) and acclimated for at least 72 h before experimentation. Mice were housed in a biosafety level 2 (BSL2) facility under specific pathogen-free conditions at 68–75 °F, with 50% humidity and a 12-h light/dark cycle.

### *Citrobacter rodentium* infections

Mice were infected by intragastric gavage with the indicated strains or mutant populations. Before inoculation, mice were deprived of food for 3–5 h. Mice were then mildly sedated by inhalation of isoflurane, and 100 μl of *C. rodentium* resuspended in PBS at the indicated dose were delivered into the stomach with a sterile feeding needle (Cadence Science). The dose was determined retrospectively by serial dilution and plating.

The *C. rodentium* population was monitored by collecting faeces from infected animals. Faeces were weighed, resuspended in sterile PBS, and homogenized using 3.2-mm stainless-steel beads and a bead beater (BioSpec Products). The concentration of *C. rodentium* was determined by serial dilution. Tn-seq libraries were expanded from faeces plated on solid LB containing streptomycin and/or kanamycin, and the bacteria were frozen for sample processing.

### Tn-seq of *C. rodentium* during infection

Inducible and traditional mutant populations were prepared by collecting ~$5 \times 10^5$ kanamycin-resistant colonies from the conjugation of *C. rodentium* to MFD*pir* + pTn with (inducible) or without (traditional) the Tn7 helper strain MFD*pir* + pJMP1039. The inducible population was cultured with 0.2% glucose before and following infection to limit the number of mutants arising outside of the animal. Aliquots of the populations were frozen at −80 °C in 20% glycerol. Immediately before infection, the populations were thawed and expanded for 3 h in LB at 37 °C with shaking at 250 r.p.m. The bacteria were then pelleted, resuspended in PBS, and administered to the animals by intragastric gavage.

For InducTn-seq, the animal's drinking water was replaced from days 3 to 8 with water containing 5% L-arabinose (Sigma Aldrich). InducTn-seq samples were collected 8 days post-inoculation by plating fresh faeces on solid LB containing 50 μg ml⁻¹ kanamycin, 200 μg ml⁻¹ streptomycin and 0.2% glucose. For traditional Tn-seq, the animals' drinking water was not supplemented with arabinose, and Tn-seq samples were collected 5 days post-inoculation by plating faeces on solid LB containing 50 μg ml⁻¹ kanamycin and 200 μg ml⁻¹ streptomycin.

### Constructing mutant strains of *C. rodentium*

In-frame deletions in *C. rodentium* were constructed using the allelic exchange protocol from ref. 96. Primers used for construction are listed in Supplementary Table 8. We linearized pTOX5 (GenBank MK972845) with the restriction enzyme SwaI, used PCR to amplify ~1 kb of the *C. rodentium* genome up- and downstream of the respective gene/locus (including two or three codons on each end), and assembled these three fragments with the HiFi DNA Assembly Master Mix. This construct was electroporated into MFD*pir*, checked by PCR, and conjugated into a streptomycin-resistant strain of *C. rodentium*. Transconjugants were selected and purified by plating consecutively three times on solid LB containing 200 μg ml⁻¹ streptomycin, 20 μg ml⁻¹ chloramphenicol and 0.2% glucose. Single colonies were expanded in liquid LB without selection for 1 h, and counter selection was performed on solid LB containing 200 μg ml⁻¹ streptomycin and 2% rhamnose. The identity of the mutant strains was confirmed by selective plating, PCR and whole-genome sequencing.

### Toxin induction experiments

The putative Cascade-repressed toxin and its variants were assembled downstream of the PBAD promoter on a replicative plasmid and electroporated directly into electrocompetent *C. rodentium*. Transformants were recovered on solid LB containing 20 μg ml⁻¹ gentamicin and 0.2% glucose. Individual colonies were picked from the transformation plates for growth rate measurements. Growth was measured by OD$_{600}$ in a 96-well plate using an Epoch2 microplate reader (BioTek). Strains were grown at 37 °C with continuous linear shaking in 200 μl of LB supplemented with 20 μg ml⁻¹ gentamicin and either glucose or arabinose at the indicated concentration.

### Reporting summary

Further information on research design is available in the Nature Portfolio Reporting Summary linked to this article.

## Data availability

The data that support the findings of this study are provided as Supplementary Tables. The sequencing reads are deposited in the Sequencing Read Archive (SRA) under accession no. PRJNA1113708. The following genomic reference sequences were used for read mapping: *E. coli* MG1655 (NC_000913.3), *E. coli* H10407 (NC_017633.1), *S. flexneri* 2457T (NC_004741.1), *S. enterica* SL1344 (FQ312003.1) and *C. rodentium* ICC168 (FN543502.1). pTn donor and pCre are deposited on Addgene (Addgene ID 236211 and 236212, respectively).

## Code availability

Graphics and figures were prepared with BioRender.com, GraphPad Prism and PowerPoint. Custom Tn-seq analysis scripts are deposited online at https://github.com/dbasta27/InducTn-seq.

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

## Acknowledgements

We thank members of the Waldor laboratory for helpful discussions and feedback on the paper. This work was supported by NIH grant no. R01AI042347 (M.K.W.), the Howard Hughes Medical Institute (HHMI; M.K.W.), NIH fellowships F31AI156949 (K.H.) and T32DK007477-37 (I.W.C.), and NIH pilot grant no. P30DK034854 (I.W.C.). This article is subject to HHMI's Open Access to Publications policy. HHMI laboratory heads have previously granted a non-exclusive CC BY 4.0 licence to the public and a sublicensable licence to HHMI in their research articles. Pursuant to those licences, the author-accepted manuscript of this article can be made freely available under a CC BY 4.0 licence immediately upon publication.

## Author contributions

D.W.B., I.W.C. and M.K.W. conceived and designed the experiments. D.W.B., I.W.C., E.J.S., J.A.H. and M.G. performed the experiments. D.W.B., I.W.C. and K.H. analysed the data. D.W.B., I.W.C. and M.K.W. wrote the paper.

## Competing interests

The authors declare no competing interests.

## Additional information

**Extended data** is available for this paper at https://doi.org/10.1038/s41564-025-01975-z.

**Correspondence and requests for materials** should be addressed to David W. Basta or Matthew K. Waldor.

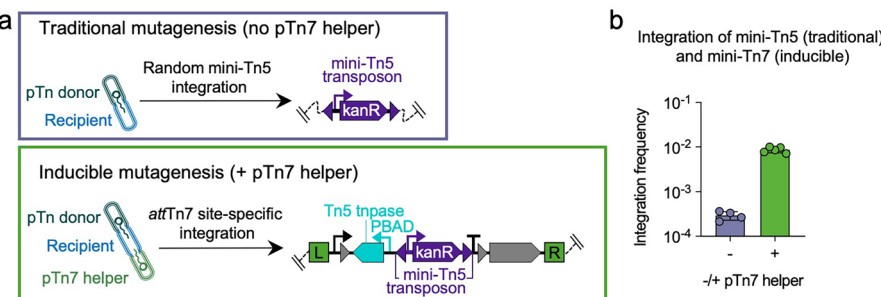

**Extended Data Fig. 1 | Comparison of mini-Tn5 and mini-Tn7 integration frequencies. a**, The non-replicative pTn donor plasmid can be used to create a traditional transposon mutant population through a one-time, random mini-Tn5 transposon integration, mediated by the Tn5 transposase, immediately after the plasmid is introduced into recipient cells. Alternatively, it can be used to create an inducible population by co-introduction of the Tn7 helper plasmid (expressing the Tn7 integration machinery), wherein transposon mutants are generated following the site-specific integration of the Tn5 transposition complex at the *att*Tn7 site in the genome. **b**, Integration at the *att*Tn7 site or random mini-Tn5

integration both result in kanamycin resistance. However, integration is ~30-fold more efficient when the Tn7 helper plasmid is present, indicating that most kanamycin-resistant colonies (total transposon integrants) represent cells containing the Tn5 transposition complex at the *att*Tn7 site rather than random mini-Tn5 mutants. The integration frequency is expressed as the ratio of kanR c.f.u to total c.f.u. The columns represent means, the error bars represent standard deviation, and the points represent biological replicates (n = 4 without pTn7 helper and n = 5 with pTn7 helper).

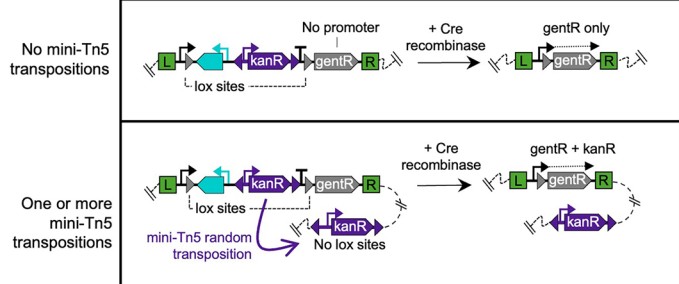

Cre-based system for detecting mini-Tn5 transposition

**Extended Data Fig. 2 | A Cre recombinase-based indicator of transposition frequency at the population level.** Following outgrowth of *att*Tn7 site-specific integrants, an optional second conjugation step can be performed to determine the population-level frequency of mini-Tn5 transposition out of the *att*Tn7 site. When a plasmid expressing the Cre recombinase is introduced into the mutant population, Cre expression leads to excision of the Tn5 transposition complex at the *att*Tn7 site via recombination of the lox sequences. Cre excision causes recipient cells to simultaneously lose the *att*Tn7-site kanamycin marker and activate expression of the gentamicin marker. Cells that did not undergo mini-Tn5 transposition prior to Cre excision of the Tn5 transposition complex become solely resistant to gentamicin, while cells that did undergo transposition retain a copy of the mini-Tn5 transposon at a random genomic location outside of the mini-Tn7 lox sites, rendering them resistant to both kanamycin and gentamicin. The ratio of gentR+kanR to gentR colonies provides a measure of transposition frequency in cells where the Tn5 transposition complex was initially integrated at the *att*Tn7 site (Fig. 2b).

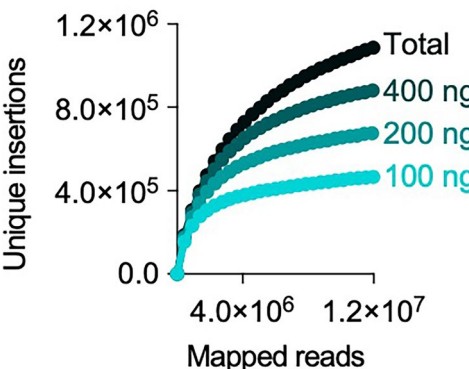

**Extended Data Fig. 3 | Total mutant diversity depends on the amount of DNA sampled.** Increasing the amount of template DNA used in amplification of the InducTn-seq library increases the number of unique insertions detected. 'Total' represents the sum of the 100, 200, and 400 ng samples. The 100 ng sample in this figure is the same as 'Induced' in Fig. 2c.

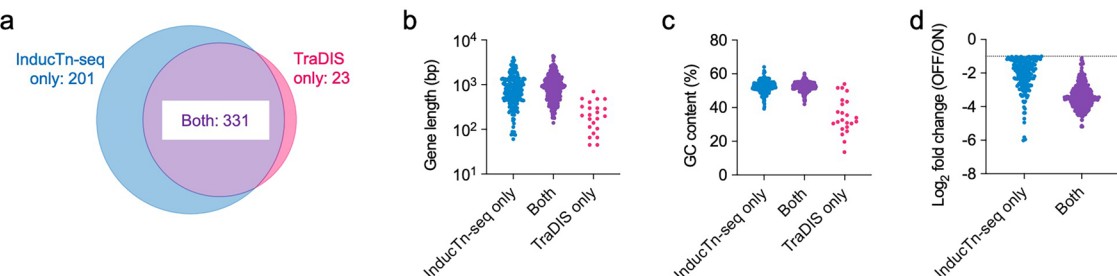

**Extended Data Fig. 4 | Comparison of InducTn-seq to traditional Tn-seq. a**, Venn diagram of *E. coli* MG1655 genes classified as having a fitness defect by InducTn-seq ($\log_2$ fold change < −1 and *P* value < 0.01 of a two-sided Mann-Whitney U test adjusted for multiple comparisons with the Benjamini-Hochberg correction) and *E. coli* BW25113 genes classified as 'essential' by TraDIS. 331 genes were commonly identified between the two screens, representing a 93.5% overlap. The discordant genes identified only by TraDIS were on average (**b**) shorter and (**c**) more AT rich. **d**, Genes exclusively identified by InducTn-seq generally had weaker fitness defects than genes concordantly identified by the two screens, suggesting that these genes may have subtle growth rate defects but are not strictly essential for growth.

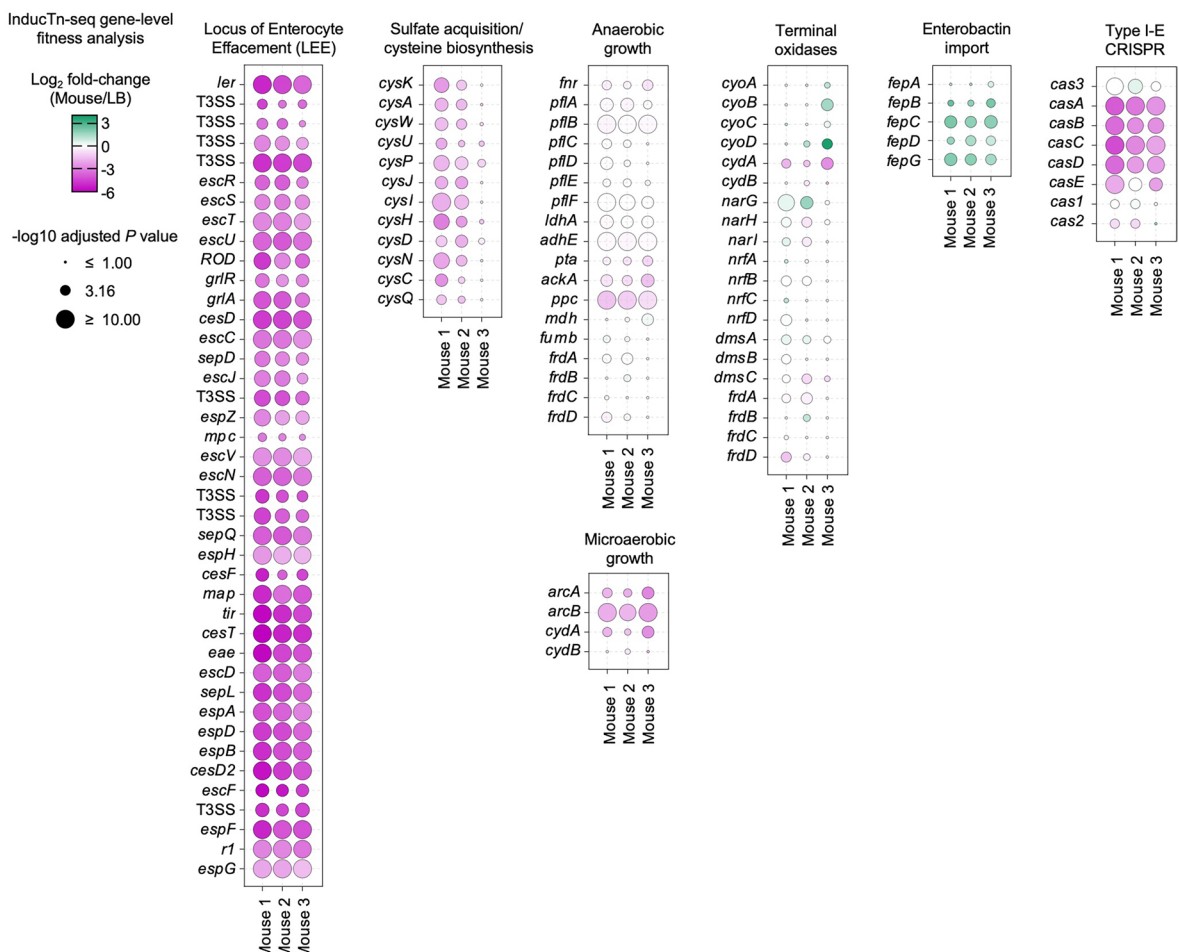

**Extended Data Fig. 5 | InducTn-seq gene-level fitness analysis of *C. rodentium* enteric colonization.** Extended results of select genes from Fig. 5g are shown for each animal replicate. *C. rodentium* mutants induced during infection and then expanded on solid LB are compared to mutants induced in culture and then expanded on solid LB. Log₂ fold-change and the two sided, Benjamini-Hochberg corrected, −log10 Mann-Whitney U *P* value is shown.

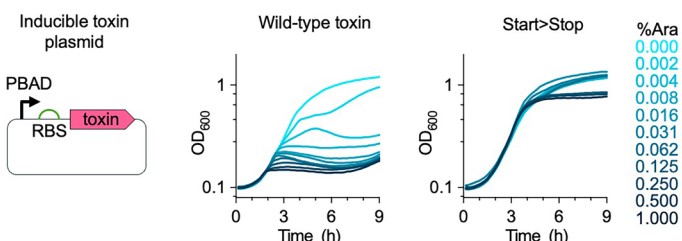

**Extended Data Fig. 6 | Toxin expression is titratable.** *C. rodentium* growth with a plasmid-borne copy of the wild-type toxin or its nonsense variant under the control of the arabinose-responsive PBAD promoter. Arabinose induction increases toxin activity in a dose-dependent manner. The average growth curve of 2-3 biological replicates is shown.

# Reporting Summary

## Statistics

For all statistical analyses, confirm that the following items are present in the figure legend, table legend, main text, or Methods section.

| n/a | Confirmed | |
|---|---|---|
| ☐ | ☒ | The exact sample size (*n*) for each experimental group/condition, given as a discrete number and unit of measurement |
| ☐ | ☒ | A statement on whether measurements were taken from distinct samples or whether the same sample was measured repeatedly |
| ☐ | ☒ | The statistical test(s) used AND whether they are one- or two-sided<br>*Only common tests should be described solely by name; describe more complex techniques in the Methods section.* |
| ☒ | ☐ | A description of all covariates tested |
| ☐ | ☒ | A description of any assumptions or corrections, such as tests of normality and adjustment for multiple comparisons |
| ☐ | ☒ | A full description of the statistical parameters including central tendency (e.g. means) or other basic estimates (e.g. regression coefficient) AND variation (e.g. standard deviation) or associated estimates of uncertainty (e.g. confidence intervals) |
| ☒ | ☐ | For null hypothesis testing, the test statistic (e.g. *F*, *t*, *r*) with confidence intervals, effect sizes, degrees of freedom and *P* value noted<br>*Give P values as exact values whenever suitable.* |
| ☒ | ☐ | For Bayesian analysis, information on the choice of priors and Markov chain Monte Carlo settings |
| ☒ | ☐ | For hierarchical and complex designs, identification of the appropriate level for tests and full reporting of outcomes |
| ☒ | ☐ | Estimates of effect sizes (e.g. Cohen's *d*, Pearson's *r*), indicating how they were calculated |

*Our web collection on statistics for biologists contains articles on many of the points above.*

## Software and code

Policy information about availability of computer code

| Data collection | For Tn-seq, data was collected using an Illumina NextSeq and FASTQ files were generated by Illumina's proprietary analysis pipeline. |
|---|---|
| Data analysis | Data analysis was performed using R version 4.4.2, Python version 3.12, and Excel. Sequencing reads were mapped using the Burrows-Wheeler Aligner (BWA) version 0.7.17-r1188. Graphics and figures were prepared with BioRender, GraphPad Prism, and Powerpoint. |

For manuscripts utilizing custom algorithms or software that are central to the research but not yet described in published literature, software must be made available to editors and reviewers. We strongly encourage code deposition in a community repository (e.g. GitHub). See the Nature Portfolio guidelines for submitting code & software for further information.

## Data

Policy information about availability of data

All manuscripts must include a data availability statement. This statement should provide the following information, where applicable:

- Accession codes, unique identifiers, or web links for publicly available datasets
- A description of any restrictions on data availability
- For clinical datasets or third party data, please ensure that the statement adheres to our policy

The sequencing reads are deposited in the sequencing read archive (SRA) under the accession number PRJNA1113708. The following genomic reference sequences were used for read mapping: E. coli MG1655 (NC_000913.3 ), E. coli H10407 (NC_017633.1), S. flexneri 2457T (NC_004741.1), S. enterica SL1344 (FQ312003.1), and C. rodentium ICC168 (FN543502.1).

## Research involving human participants, their data, or biological material

Policy information about studies with human participants or human data. See also policy information about sex, gender (identity/presentation), and sexual orientation and race, ethnicity and racism.

| | |
|---|---|
| Reporting on sex and gender | N/A |
| Reporting on race, ethnicity, or other socially relevant groupings | N/A |
| Population characteristics | N/A |
| Recruitment | N/A |
| Ethics oversight | N/A |

Note that full information on the approval of the study protocol must also be provided in the manuscript.

# Field-specific reporting

Please select the one below that is the best fit for your research. If you are not sure, read the appropriate sections before making your selection.

☒ Life sciences ☐ Behavioural & social sciences ☐ Ecological, evolutionary & environmental sciences

For a reference copy of the document with all sections, see nature.com/documents/nr-reporting-summary-flat.pdf

# Life sciences study design

All studies must disclose on these points even when the disclosure is negative.

| | |
|---|---|
| Sample size | Sample sizes were determined by the variation observed during preliminary experiments. CFU measurements are reported on a log10 scale. |
| Data exclusions | No data were excluded from the analyses |
| Replication | All experiments were repeated a minimum of two independent times. All attempts at replication were successful. |
| Randomization | Mice were randomly assigned to experimental groups |
| Blinding | Blinding was not used in these studies because the quantifiable data (CFUs, sequencing reads) are not subject to investigator bias |

# Reporting for specific materials, systems and methods

We require information from authors about some types of materials, experimental systems and methods used in many studies. Here, indicate whether each material, system or method listed is relevant to your study. If you are not sure if a list item applies to your research, read the appropriate section before selecting a response.

### Materials & experimental systems

| n/a | Involved in the study |
|---|---|
| ☒ | Antibodies |
| ☒ | Eukaryotic cell lines |
| ☒ | Palaeontology and archaeology |
| ☐ | ☒ Animals and other organisms |
| ☒ | Clinical data |
| ☒ | Dual use research of concern |
| ☒ | Plants |

### Methods

| n/a | Involved in the study |
|---|---|
| ☒ | ChIP-seq |
| ☒ | Flow cytometry |
| ☒ | MRI-based neuroimaging |

# Animals and other research organisms

Policy information about studies involving animals; ARRIVE guidelines recommended for reporting animal research, and Sex and Gender in Research

| | |
|---|---|
| Laboratory animals | Adult (9-12 weeks), female, C57BL/6J mice were used |
| Wild animals | No wild animals were used in the study |
| Reporting on sex | All mice were female. |
| Field-collected samples | No field collected samples were used in the study |
| Ethics oversight | Animal studies were conducted at the Brigham and Women's Hospital in compliance with the Guide for the Care and Use of Laboratory Animals and according to protocols reviewed and approved by the Brigham and Women's Hospital Institutional Animal Care and Use Committee (protocol 2016N000416). |

Note that full information on the approval of the study protocol must also be provided in the manuscript.

# Plants

| | |
|---|---|
| Seed stocks | *Report on the source of all seed stocks or other plant material used. If applicable, state the seed stock centre and catalogue number. If plant specimens were collected from the field, describe the collection location, date and sampling procedures.* |
| Novel plant genotypes | *Describe the methods by which all novel plant genotypes were produced. This includes those generated by transgenic approaches, gene editing, chemical/radiation-based mutagenesis and hybridization. For transgenic lines, describe the transformation method, the number of independent lines analyzed and the generation upon which experiments were performed. For gene-edited lines, describe the editor used, the endogenous sequence targeted for editing, the targeting guide RNA sequence (if applicable) and how the editor was applied.* |
| Authentication | *Describe any authentication procedures for each seed stock used or novel genotype generated. Describe any experiments used to assess the effect of a mutation and, where applicable, how potential secondary effects (e.g. second site T-DNA insertions, mosiacism, off-target gene editing) were examined.* |

