## [Peer Review File · Nature Microbiology]

Inducible transposon mutagenesis identifies bacterial fitness determinants during infection in mice

Corresponding Author: Dr David Basta

Version 0:

Reviewer comments:

Reviewer #1

(Remarks to the Author)

This manuscript presents an inducible Tn-seq method and shows its application to a *Citrobacter*-mouse model. The method is novel and its potential to overcome limitations from pathogen population bottlenecks is exciting. However, the authors provide only minimal information about their *Citrobacter* results. At present, new biological insights remain thus very limited.

Comments:

Line 160-2: The number of unique insertions was higher than the colony number also for non-induced colonies (10^4 vs. 10^3). This demonstrates ongoing transposition after plating even in absence of inducer. This should be mentioned.

Line 156-7; 194-5: The ON-OFF transition is likely incomplete because of the leaky PBAD promoter. Thus ongoing transposition is likely to occur after the end of induction. This should be explicitly mentioned.

Line 334-5, Fig. 5a: What is the extent of background transposition in mice without inducer? Part of the transposition events are likely to have occurred after plating the faeces on agar plates? This could be analyzed by sequencing a few colonies separately.

Fig. 6b: For mouse experiments, faeces was recovered and plated on LB plates prior to sequencing. Strains that have wild-type fitness in mice but reduced fitness on the LB plates would, therefore, show still a decreased overall fitness. Thus, the label "colonization factors" is a bit misleading as the x-axis in reality shows a combination of colonization factors with fitness on LB. On the other hand, it is confusing that genes required on LB plates (and even in mice, see below) such as *sdhABCD* exhibit no fitness effect in the composite mouse-plate assay.

Fig. 6d: *arcA* has a strong early phenotype in mice, followed by weaker effects. The weaker *arcA* phenotype in InducTn could reflect the induction of transposition after the critical early infection phase?

Line 442-5: The defined *sdhABCD* mutant showed continuously increasing fitness defects in mice as well as on LB plates, while InducTn (which combines both conditions) did not show any effect. The offered explanation of an early effect before transposition is not convincing as the data in Fig. 6d show a particularly pronounced fitness disadvantage from day 11 to 17, which is actually after transposition induction was already completed.

Citrobacter infection biology: The authors present only a limited analysis of their extensive InducTn results. The oxygen-related phenotypes are mostly convincing (except *shdABCD*, see above) but also largely known or expected from previous research on intestinal pathogens. The authors should provide a thorough analysis of their large dataset with a focus on novel insights into *Citrobacter* infection biology.

Reviewer #2

(Remarks to the Author)

This work introduces "InducTn-seq," an improved method for genome-wide genetic screening in bacteria that overcomes key limitations of traditional transposon sequencing (Tn-seq). The key innovation is the use of an inducible transposase system, allowing researchers to generate diverse mutant libraries on demand, even starting from a single bacterial cell. This provides several major benefits, including improving sensitivity for detecting subtle fitness effects, the ability to quantitatively measure fitness impacts in essential genes and overcoming the "bottleneck problem" in animal infection models. The authors demonstrate these advantages by using InducTn-seq to perform a genome-wide screen in a mouse model of *Citrobacter rodentium* infection, revealing new insights into metabolic pathways important for pathogenesis. Overall, InducTn-seq greatly expands the power and applicability of bacterial genetic screening, especially for in vivo studies.

Major comments

1. Figure 5a – labeling is unclear to me. Does blue/purple represent InducTn-seq vs traditional Tn-seq or arabinose treatment? Please revise figure.
2. The authors claim that arabinose induction did not affect *Citrobacter* abundance. However, the data should include clear statistical comparisons between treated and untreated samples to support this assertion.
3. Although arabinose induction did not affect *Citrobacter* cfu, the introduction of arabinose itself could have an effect on bacterial metabolism, microbiota composition, virulence and maybe even prophage induction (see Cottam, 2024, Nat Comm; Tomioka, Cell Rep 2022; Hu, 2023, Gut Microbes). This should be discussed.
4. There is clear evidence that the *cydAB* terminal oxidase plays a central role in *Citrobacter* colonization (Lopez, Science, 2017). Were related genes identified in the screen? Regardless, this should be mentioned and discussed.

Minor comments

1. the text in lines 85-86 is hidden by figure 1.
2. Line 320 – EPEC & EHEC – full names of the pathogens should be included.
3. Lines 454-456 – Please provide a reference for extra-host lifestyle distinctions. Also, obligate anaerobe bacteria can also be pathogens (*Clostridium* species *difficile*, *tetani*, *botulinum*, *perfringens*, *Bacteroides fragilis*..), and commensal can grow aerobically (*E. coli*, *Klebsiella*..). The sentence should be phrased more accurately.

Reviewer #3

(Remarks to the Author)

Basta et al describes the development of InducTn-seq, which is a method that allows genome-wide transposon insertion mutations in an inducible manner. This system leverages initial genomic integration into an att site, and subsequent Tn5 copy-paste random genome wide insertions of the cassette, controlled with arabinose induction. The authors tested this system first in *E. coli* in cultures to demonstrate the sensitivity of the method to detect insertions into essential genes. The system was subsequently tested in different proteobacteria to show versatility. In a mouse pathogen colonization model with *C. rodentium*, the authors demonstrated the use of the system to identify gut colonization factors including those associated with oxygen-related metabolism during the course of *C. rodentium* induced enteric infection.

The work tries to address a known challenge in traditional Tn-seq system, which is that environments that bottleneck a Tn-seq population cannot be easily studied. The proposed solution can overcome this issue by using an inducible transposition system. Overall, the manuscript is clearly written and the results are adequately interpreted from well-designed experiments. The innovation of the system is rather straightforward and is likely to be useful for others in the field studying microbial colonization or pathogenesis. A few questions that the authors could clarify are as follows:

1. The kinetics of the transposition in the system. How many transpositions is expected to occur per generation? The authors describe the accumulation of multiple transpositions in Fig 2d, but it is unclear how long it took to accumulate those. Is the expectation that continual passaging of the transposon will continually accumulate even more transposons over time? A better discussion/treatment of this could be helpful.
2. In the absence of arabinose, there is still some residual transposition presumably from leaky expression of pBAD. Could the use of more stringent inducible expression system reduce the background? More importantly, how does this leaky mutagenesis affect the initial genomic background of the population in the screen? The animal to animal correlation seem to be high, but some discussion of this would be good.
3. In the *C. rodentium* model, how many mutations were accumulated per cell over time? By the 8th day, one presumes that there are multiple insertional mutations per cell yet the fitness measurements is an integration across all of those mutations. There might be some epistatic effects of multiple mutations that are being observed. Some analysis or discussions of this issue may be warranted.

Decision Letter:

19th July 2024

Dear David,

Thank you (and Matt and Ian) very much for taking the time to meet with me today, and to discuss the experimental plan. Your plans to address the technical points raised by the reviewers and to reframe the *Citrobacter* Inducible Tnseq results look very promising. The formal decision email is below, if you have any other questions during the revisions, please don't hesitate to send me an email. I'm happy to help.

Thank you for your patience while your manuscript "Inducible transposon mutagenesis for genome-scale forward genetics" was under peer-review at Nature Microbiology. It has now been seen by 3 referees, whose expertise and comments you will find at the end of this email. They find your work of some potential interest, however they have raised a number of concerns that will need to be addressed before we can consider publication of the work in Nature Microbiology.

As we discussed, the revised manuscript will need to address the referee concerns regarding quantifying the background mutagenesis, particularly during outgrowth, and quantifying the number of transposition events that occur during colonisation in vivo. The revisions will also need to address the reviewers' concerns regarding the biological insight gained from using the InducTn-seq approach.

Should further experimental data allow you to address these criticisms, we would be happy to look at a revised manuscript.

Please include a data availability statement as a separate section after Methods but before references, under the heading "Data Availability". This section should inform readers about the availability of the data used to support the conclusions of your study. This information includes accession codes to public repositories (data banks for protein, DNA or RNA sequences, microarray, proteomics data etc...), references to source data published alongside the paper, unique identifiers such as URLs to data repository entries, or data set DOIs, and any other statement about data availability. At a minimum, you should include the following statement: "The data that support the findings of this study are available from the corresponding author upon request", mentioning any restrictions on availability. If DOIs are provided, we also strongly encourage including these in the Reference list (authors, title, publisher (repository name), identifier, year). For more guidance on how to write this section please see: <http://www.nature.com/authors/policies/data/data-availability-statements-data-citations.pdf>

* If you have not done so already we suggest that you begin to revise your manuscript so that it conforms to our Article format instructions at <http://www.nature.com/nmicrobiol/info/final-submission>. Refer also to any guidelines provided in this letter.

When submitting the revised version of your manuscript, please pay close attention to our [href="https://www.nature.com/nature-portfolio/editorial-policies/image-integrity">Digital Image Integrity Guidelines](https://www.nature.com/nature-portfolio/editorial-policies/image-integrity) and to the following points below:

Link Redacted

Note: This url links to your confidential homepage and associated information about manuscripts you may have submitted or be reviewing for us. If you wish to forward this e-mail to co-authors, please delete this link to your homepage first.

Nature Microbiology is committed to improving transparency in authorship. As part of our efforts in this direction, we are now requesting that all authors identified as 'corresponding author' on published papers create and link their Open Researcher and Contributor Identifier (ORCID) with their account on the Manuscript Tracking System (MTS), prior to acceptance. This applies to primary research papers only. ORCID helps the scientific community achieve unambiguous attribution of all scholarly contributions. You can create and link your ORCID from the home page of the MTS by clicking on 'Modify my Springer Nature account'. For more information please visit [please visit www.springernature.com/orcid](http://www.springernature.com/orcid).

If you wish to submit a suitably revised manuscript we would hope to receive it within 6 months. If you cannot send it within this time, please let us know. We will be happy to consider your revision, even if a similar study has been accepted for publication at Nature Microbiology or published elsewhere (up to a maximum of 6 months).

Yours sincerely,

Reviewer Comments:

Reviewer #1 (Remarks to the Author):

This manuscript presents an inducible Tn-seq method and shows its application to a *Citrobacter*-mouse model. The method is novel and its potential to overcome limitations from pathogen population bottlenecks is exciting. However, the authors provide only minimal information about their *Citrobacter* results. At present, new biological insights remain thus very limited.

Comments:

Line 160-2: The number of unique insertions was higher than the colony number also for non-induced colonies (10^4 vs. 10^3). This demonstrates ongoing transposition after plating even in absence of inducer. This should be mentioned.

Line 156-7; 194-5: The ON-OFF transition is likely incomplete because of the leaky PBAD promoter. Thus ongoing transposition is likely to occur after the end of induction. This should be explicitly mentioned.

Line 334-5, Fig. 5a: What is the extent of background transposition in mice without inducer? Part of the transposition events are likely to have occurred after plating the faeces on agar plates? This could be analyzed by sequencing a few colonies separately.

Fig. 6b: For mouse experiments, faeces was recovered and plated on LB plates prior to sequencing. Strains that have wild-type fitness in mice but reduced fitness on the LB plates would, therefore, show still a decreased overall fitness. Thus, the label "colonization factors" is a bit misleading as the x-axis in reality shows a combination of colonization factors with fitness on LB. On the other hand, it is confusing that genes required on LB plates (and even in mice, see below) such as *sdhABCD* exhibit no fitness effect in the composite mouse-plate assay.

Fig. 6d: *arcA* has a strong early phenotype in mice, followed by weaker effects. The weaker *arcA* phenotype in InducTn could reflect the induction of transposition after the critical early infection phase?

Line 442-5: The defined *sdhABCD* mutant showed continuously increasing fitness defects in mice as well as on LB plates, while InducTn (which combines both conditions) did not show any effect. The offered explanation of an early effect before transposition is not convincing as the data in Fig. 6d show a particularly pronounced fitness disadvantage from day 11 to 17, which is actually after transposition induction was already completed.

Citrobacter infection biology: The authors present only a limited analysis of their extensive InducTn results. The oxygen-related phenotypes are mostly convincing (except *shdABCD*, see above) but also largely known or expected from previous research on intestinal pathogens. The authors should provide a thorough analysis of their large dataset with a focus on novel insights into *Citrobacter* infection biology.

Reviewer #2 (Remarks to the Author):

This work introduces "InducTn-seq," an improved method for genome-wide genetic screening in bacteria that overcomes key limitations of traditional transposon sequencing (Tn-seq). The key innovation is the use of an inducible transposase system, allowing researchers to generate diverse mutant libraries on demand, even starting from a single bacterial cell. This provides several major benefits, including improving sensitivity for detecting subtle fitness effects, the ability to quantitatively measure fitness impacts in essential genes and overcoming the "bottleneck problem" in animal infection models. The authors demonstrate these advantages by using InducTn-seq to perform a genome-wide screen in a mouse model of *Citrobacter rodentium* infection, revealing new insights into metabolic pathways important for pathogenesis. Overall, InducTn-seq greatly expands the power and applicability of bacterial genetic screening, especially for in vivo studies.

Major comments

1. Figure 5a – labeling is unclear to me. Does blue/purple represent inducTn-seq vs traditional Tn-seq or arabinose treatment? Please revise figure.
2. The authors claim that arabinose induction did not affect *Citrobacter* abundance. However, the data should include clear statistical comparisons between treated and untreated samples to support this assertion.
3. Although arabinose induction did not affect *Citrobacter* cfu, the introduction of arabinose itself could have an effect on bacterial metabolism, microbiota composition, virulence and maybe even prophage induction (see Cottam, 2024, Nat Comm; Tomioka, Cell Rep 2022; Hu, 2023, Gut Microbes). This should be discussed.
4. There is clear evidence that the *cydAB* terminal oxidase plays a central role in *Citrobacter* colonization (Lopez, Science, 2017). Were related genes identified in the screen? Regardless, this should be mentioned and discussed.

Minor comments

1. the text in lines 85-86 is hidden by figure 1.
2. Line 320 – EPEC & EHEC – full names of the pathogens should be included.
3. Lines 454-456 – Please provide a reference for extra-host lifestyle distinctions. Also, obligate anaerobe bacteria can also be pathogens (*Clostridium* species *difficile*, *tetani*, *botulinum*, *perfringens*, *Bacteroides fragilis*..), and commensal can grow aerobically (*E. coli*, *Klebsiella*..). The sentence should be phrased more accurately.

Reviewer #3 (Remarks to the Author):

Basta et al describes the development of InducTn-seq, which is a method that allows genome-wide transposon insertion mutations in an inducible manner. This system leverages initial genomic integration into an att site, and subsequent Tn5 copy-paste random genome wide insertions of the cassette, controlled with arabinose induction. The authors tested this system first in *E. coli* in cultures to demonstrate the sensitivity of the method to detect insertions into essential genes. The system was subsequently tested in different proteobacteria to show versatility. In a mouse pathogen colonization model with *C. rodentium*, the authors demonstrated the use of the system to identify gut colonization factors including those associated with oxygen-related metabolism during the course of *C. rodentium* induced enteric infection.

The work tries to address a known challenge in traditional Tn-seq system, which is that environments that bottleneck a Tn-seq population cannot be easily studied. The proposed solution can overcome this issue by using an inducible transposition system. Overall, the manuscript is clearly written and the results are adequately interpreted from well-designed experiments. The innovation of the system is rather straightforward and is likely to be useful for others in the field studying microbial colonization or pathogenesis. A few questions that the authors could clarify are as follows:

1. The kinetics of the transposition in the system. How many transpositions is expected to occur per generation? The authors describe the accumulation of multiple transpositions in Fig 2d, but it is unclear how long it took to accumulate those. Is the expectation that continual passaging of the transposon will continually accumulate even more transposons over time? A better discussion/treatment of this could be helpful.
2. In the absence of arabinose, there is still some residual transposition presumably from leaky expression of pBAD. Could the use of more stringent inducible expression system reduce the background? More importantly, how does this leaky mutagenesis affect the initial genomic background of the population in the screen? The animal to animal correlation seem to be high, but some discussion of this would be good.
3. In the *C. rodentium* model, how many mutations were accumulated per cell over time? By the 8th day, one presumes that there are multiple insertional mutations per cell yet the fitness measurements is an integration across all of those mutations. There might be some epistatic effects of multiple mutations that are being observed. Some analysis or discussions of this issue may be warranted.

Version 1:

Reviewer comments:

Reviewer #1

(Remarks to the Author)

The authors have adequately addressed all my comments and the newly included insights into *Citrobacter* biology of a cryptic CRISPR-repressed toxin are interesting. I have no further comments.

Reviewer #3

(Remarks to the Author)

The authors adequately responded to all of the reviewers concerns in the revision including new data and text. The reviewer has no further issues.

Reviewer #4

(Remarks to the Author)

Authors have successfully addressed all my original points.

My only additional comment has to do with the removed barcoded competition assays. From my point of view, these data should be further analyzed/discussed in the paper according to comments raised by reviewer 1. The identification and characterization of the toxin-antitoxin system within the CRISPR locus is novel and exciting, but the barcoded competition experiments are important for the validation and/or discussion of the genomic (InducTn-seq) data. Therefore, I recommend both datasets to be included in the manuscript.

Decision Letter:

Our ref: NMICROBIOL-24061666A

3rd February 2025

Dear David,

Thank you for submitting your revised manuscript "Inducible transposon mutagenesis for genome-scale forward genetics" (NMICROBIOL-24061666A). Thank you also for your patience during this second round of review. The manuscript has now been seen by two of the original referees plus an additional reviewer who was brought in as a replacement. Their comments are below, and as you can see, the reviewers find that the paper has improved in revision. We'll be happy in principle to publish it in Nature Microbiology, pending minor revisions to satisfy the referees' final requests and to comply with our editorial and formatting guidelines.

As you'll see from the comments, Reviewer #4 asked that the barcoding data that was taken out during revision be included and the points made by Reviewer #1 be discussed. From our editorial perspective, this is optional. We agree that including it validates some findings and there are some useful points to discuss, but as Reviewer #1 already finds that their concerns have been addressed, we will leave this to your discretion. Our next steps will be to do our pre-accept checks on the manuscript and figures, so please don't make any changes or work on the files as yet. We will send you a checklist detailing our editorial and formatting requirements in about a week. Please do not upload the final materials and make any revisions until you receive this additional information from us. This is to avoid too many rounds of reformatting and hopefully keep it to one final step.

Thank you again for your interest in Nature Microbiology Please do not hesitate to contact me if you have any questions.

Best wishes,

Reviewer #1 (Remarks to the Author):

The authors have adequately addressed all my comments and the newly included insights into *Citrobacter* biology of a cryptic CRISPR-repressed toxin are interesting. I have no further comments.

Reviewer #2 (withdrew)

Reviewer #3 (Remarks to the Author):

The authors adequately responded to all of the reviewers concerns in the revision including new data and text. The reviewer has no further issues.

Reviewer #4 (Remarks to the Author):

Authors have successfully addressed all my original points.

My only additional comment has to do with the removed barcoded competition assays. From my point of view, these data should be further analyzed/discussed in the paper according to comments raised by reviewer 1. The identification and characterization of the toxin-antitoxin system within the CRISPR locus is novel and exciting, but the barcoded competition experiments are important for the validation and/or discussion of the genomic (InducTn-seq) data. Therefore, I recommend both datasets to be included in the manuscript.

Version 2:

Decision Letter:

3rd March 2025

Dear Dr Basta,

I am pleased to accept your Article "Inducible transposon mutagenesis identifies bacterial fitness determinants during infection in mice" for publication in Nature Microbiology. Thank you for having chosen to submit your work to us and many congratulations.

Authors may need to take specific actions to achieve [compliance](https://www.springernature.com/gp/open-research/funding/policy-compliance-faqs) with funder and institutional open access mandates. If your research is supported by a funder that requires immediate open access (e.g. according to [Plan S principles](https://www.springernature.com/gp/open-research/plan-s-compliance)) then you should select the gold OA route, and we will direct you to the compliant route where possible. For authors selecting the subscription publication route, the journal's standard licensing terms will need to be accepted, including [self-archiving policies](https://www.nature.com/nature-portfolio/editorial-policies/self-archiving-and-license-to-publish). Those licensing terms will supersede any other terms that the author or any third party may assert apply to any version of the manuscript.

With kind regards,

P.S. Click on the following link if you would like to recommend Nature Microbiology to your librarian
<http://www.nature.com/subscriptions/recommend.html#forms>

** Visit the Springer Nature Editorial and Publishing website at http://editorial-jobs.springernature.com?utm_source=ejP_NMicro_email&utm_medium=ejP_NMicro_email&utm_campaign=ejp_NMicro for more information about our career opportunities. If you have any questions please click [here](mailto:editorial.publishing.jobs@springernature.com).**

Response to Referees

Referee #1 (review in black, response in blue)

This manuscript presents an inducible Tn-seq method and shows its application to a *Citrobacter*-mouse model. The method is novel and its potential to overcome limitations from pathogen population bottlenecks is exciting. However, the authors provide only minimal information about their *Citrobacter* results. At present, new biological insights remain thus very limited.

We thank the Referee for their positive comments on the manuscript and agree that the results of our genetic screen warranted more careful treatment and a more thorough analysis. Therefore, we have created a new section in the Results entitled “*InducTn-seq identifies C. rodentium* colonization factors” (lines 389-438 in the revised manuscript), where we perform a detailed analysis of our *in vivo* dataset. This new analysis is reflected in a revised Fig. 5, specifically in panels g and h, and a new Extended Data Fig 5. Importantly, our results align with previously identified *C. rodentium* colonization factors, including most genes within the LEE pathogenicity island, multiple amino acid biosynthetic pathways, and energy-conserving metabolic pathways active under low oxygen conditions. We have also integrated our previous discussion of the CyoABCDE complex into a broader discussion on the role of different terminal oxidases during infection, highlighting the metabolic constraints on *C. rodentium* within the murine host.

Importantly, our new analyses have led to the surprising discovery that the Type I-E CRISPR system of *C. rodentium* is required for intestinal colonization. In the revision, we devote a new Fig. 6 and Extended Data Fig. 6 to identifying why CRISPR is required for growth within the mouse. We found that genes encoding the CRISPR RNA-binding Cascade complex (*casABCDE*) were among the top hits in our screen, but unexpectedly appeared to play a role beyond phage defense, since the associated Cas3 nuclease was dispensable during infection. To explain these results, we demonstrate that Cascade activity is necessary to repress a cryptic toxin activated within the mouse. This discovery represents the first example of a toxin-antitoxin system that conditionally addicts a bacterium to its CRISPR locus during infection.

Comments:

Line 160-2: The number of unique insertions was higher than the colony number also for non-induced colonies (10^4 vs. 10^3). This demonstrates ongoing transposition after plating even in absence of inducer. This should be mentioned.

Thank you for the comment. In the revised manuscript we now state on lines 157-159, “In the absence of induction, $\sim 1.3 \times 10^4$ unique insertions were detected, which we attribute to a low-level of transposition in the absence of inducer (i.e., leakiness of the PBAD promoter).”

Line 156-7; 194-5: The ON-OFF transition is likely incomplete because of the leaky PBAD promoter. Thus ongoing transposition is likely to occur after the end of induction. This should be explicitly mentioned.

We agree that promoter leakiness is an important consideration and thank the Referee for identifying this area for further clarification. We have added additional language to the revised manuscript,

including the lines 157-159 referenced above; lines 198-200: “We note that ongoing low-level transposition appears to set a lower bound on the magnitude of insertion depletion (Fig. 3b), resulting in an equilibrium between selection and background transposition”; and lines 643-645 in the Discussion: “In all cases, tight promoter regulation is important to ensure that in the absence of inducer the effects of selection are not obscured by ongoing background mutagenesis”.

We appreciate the Referee’s concern that background transposition could potentially confound interpretation of our results. However, empirically this does not appear to be an issue. Our results in Fig. 2b indicate that when cells are grown without induction less than 1% of the total cells are mutants compared to ~28% with induction. This low level of background transposition does not significantly affect phenotypic interpretation of our Tn-seq data as demonstrated by the analysis in the sections “*Insertions in essential genes*” and “*A sensitive, streamlined framework for mutant fitness analysis*” (lines 182-271). In these sections, we observe that the background level of transposition sets a lower bound observable as a plateau in the \log_2 fold-change. Mutants in genes required for LB growth are depleted from the population but do not completely disappear, instead reaching an equilibrium between depletion and background transposition (Fig. 3b). Critically, the rate of background mutagenesis does not outweigh the rate at which insertions are depleted from the population, since we observe up to a 64-fold reduction in transposon insertion frequency when comparing an induced population (ON) to the same population outgrown without induction (OFF) (Fig. 3c).

Line 334-5, Fig. 5a: What is the extent of background transposition in mice without inducer? Part of the transposition events are likely to have occurred after plating the faeces on agar plates? This could be analyzed by sequencing a few colonies separately.

The Referee raises an important issue. To answer these questions we have performed additional experiments to measure *C. rodentium* transposition frequency in culture and during infection with and without the addition of arabinose. We present these results in the new Fig. 5b and describe them on lines 357-368 of the revised manuscript.

We found that background transposition in animals in the absence of arabinose supplementation was higher than in culture (compare mouse minus arabinose to culture minus arabinose in Fig. 5b), likely due to the presence of arabinose in the mouse chow. However, transposition during infection remains responsive to arabinose induction (compare mouse plus arabinose to mouse minus arabinose).

To assess the impact of new mutants arising during outgrowth of the fecal population, we plated the uninduced animal inoculum directly onto LB agar plates without arabinose and measured the number of new *C. rodentium* mutants. This experiment serves as a proxy for the contribution of new transposition events following plating of the feces on agar. Our results indicate that few new mutants are generated when *C. rodentium* from the feces of infected mice is expanded on LB (compare inoculum to culture minus arabinose in Fig. 5b).

These new experiments collectively demonstrate that the mutants observed in the fecal population primarily arise due to arabinose induction within the mouse rather than during expansion on LB culture plates in the absence of arabinose (compare ~42% mutants in mouse plus arabinose to ~2% mutants in culture minus arabinose in Fig. 5b).

Furthermore, we can empirically conclude that background transposition is not confounding our results because we are able to detect colonization-specific factors such as genes within the LEE pathogenicity island (Fig. 5g). LEE genes contribute to *C. rodentium* fitness during colonization but not during culture. Therefore, if significant numbers of new mutants arise in LEE genes during outgrowth, we would not be able to identify the LEE genes as colonization factors.

Fig. 6b: For mouse experiments, faeces was recovered and plated on LB plates prior to sequencing. Strains that have wild-type fitness in mice but reduced fitness on the LB plates would, therefore, show still a decreased overall fitness. Thus, the label “colonization factors” is a bit misleading as the x-axis in reality shows a combination of colonization factors with fitness on LB. On the other hand, it is confusing that genes required on LB plates (and even in mice, see below) such as *sdhABCD* exhibit no fitness effect in the composite mouse-plate assay.

The Referee’s point is well taken; based on our experimental design our original interpretation of “colonization factors” incorrectly included LB-specific factors. We have modified our analysis by comparing animal-derived samples expanded on LB without arabinose (mouse) to a population induced in culture and then expanded on LB without arabinose (OFF). This new comparison more accurately identifies colonization-specific factors by comparing two populations both expanded similarly on LB, thereby controlling for the contribution of LB growth in the determination of mouse fitness. The results of this new comparison are shown in the new Fig. 5g,h and Extended Data Fig. 5. The phenotype of mutants in *sdhABCD* is discussed below.

Fig. 6d: *arcA* has a strong early phenotype in mice, followed by weaker effects. The weaker *arcA* phenotype in InducTn could reflect the induction of transposition after the critical early infection phase?

We have removed the barcoded competition experiments and replaced them with an in-depth characterization of a novel toxin-antitoxin system within the CRISPR locus of *C. rodentium*. However, the Referee is likely correct regarding the critical early period. We attribute the difference in fold change between the two experiments to the fact that mutagenesis was induced from days 3 to 8 in the screen and the population was measured on day 8, while the barcoded mutants were competed throughout the entire course of the infection and samples were collected periodically from days 2 to 17. If we examine the period of the barcoded competition that best matches with the InducTn-seq experiment (days 3-8), the *arcA* mutant had a \log_2 fold-change of approximately -3.5 in the competition compared to -3.4 in the original screen.

Line 442-5: The defined *sdhABCD* mutant showed continuously increasing fitness defects in mice as well as on LB plates, while InducTn (which combines both conditions) did not show any effect. The offered explanation of an early effect before transposition is not convincing as the data in Fig. 6d show a particularly pronounced fitness disadvantage from day 11 to 17, which is actually after transposition induction was already completed.

We consider the relative phenotypic agreement from days 3-8 between the InducTn-seq screen and all of the mutants in the barcoded competition as validation of our screen (i.e., an *sdhABCD* mutant had a phenotype between an *fnr* and *cyoABCDE* mutant in both experiments, while the *ackA* and *arcA* mutants had stronger defects).

The Referee correctly identifies the fitness defect of the *sdhABCD* deletion strain observed from days 11-17 in the barcode competition. However, it is important to note that our InducTn-seq screen was not designed to capture fitness changes occurring at this later stage of infection. We are intrigued by the potential of InducTn-seq to identify temporal fitness requirements during infection and are actively investigating this in ongoing experiments.

Citrobacter infection biology: The authors present only a limited analysis of their extensive InducTn results. The oxygen-related phenotypes are mostly convincing (except *shdABCD*, see above) but also largely known or expected from previous research on intestinal pathogens. The authors should provide a thorough analysis of their large dataset with a focus on novel insights into Citrobacter infection biology.

As described in detail above, we have now performed a thorough analysis of our InducTn-seq dataset and characterized a novel role for CRISPR in *C. rodentium* infection biology. We hope this additional work satisfies the Referee's concerns.

Referee #2 (review in black, response in blue)

This work introduces "InducTn-seq," an improved method for genome-wide genetic screening in bacteria that overcomes key limitations of traditional transposon sequencing (Tn-seq). The key innovation is the use of an inducible transposase system, allowing researchers to generate diverse mutant libraries on demand, even starting from a single bacterial cell. This provides several major benefits, including improving sensitivity for detecting subtle fitness effects, the ability to quantitatively measure fitness impacts in essential genes and overcoming the "bottleneck problem" in animal infection models. The authors demonstrate these advantages by using InducTn-seq to perform a genome-wide screen in a mouse model of Citrobacter rodentium infection, revealing new insights into metabolic pathways important for pathogenesis. Overall, InducTn-seq greatly expands the power and applicability of bacterial genetic screening, especially for in vivo studies.

We thank the Referee for their positive comments on the manuscript.

Major comments

1. Figure 5a – labeling is unclear to me. Does blue/purple represent inducTn-seq vs traditional Tn-seq or arabinose treatment? Please revise figure.

We apologize for this oversight and thank the Referee for identifying the ambiguity in our labeling. We have created more specific labels in the updated Fig. 5a ("Traditional Tn-seq," "InducTn-seq (uninduced)," and "InducTn-seq + L-ara d3-8").

2. The authors claim that arabinose induction did not affect Citrobacter abundance. However, the data should include clear statistical comparisons between treated and untreated samples to support this assertion.

We have performed a statistical comparison between treated and untreated samples and included this information in the Fig. 5 legend of the revised manuscript.

3. Although arabinose induction did not affect *Citrobacter* cfu, the introduction of arabinose itself could have an effect on bacterial metabolism, microbiota composition, virulence and maybe even prophage induction (see Cottam, 2024, Nat Comm; Tomioka, Cell Rep 2022; Hu, 2023, Gut Microbes). This should be discussed.

We thank the Referee for raising this important point. We now raise this point, and include the provided references in the Discussion on lines 635-638 of the revised manuscript: “Although we used the arabinose-responsive PBAD promoter to induce transposase expression, we note that arabinose supplementation during infection experiments may have unintended effects on virulence, as well as on both microbial and host metabolism”.

4. There is clear evidence that the *cydAB* terminal oxidase plays a central role in *Citrobacter* colonization (Lopez, Science, 2017). Were related genes identified in the screen? Regardless, this should be mentioned and discussed.

We have added additional discussion of this point in a new section of the Results entitled “*InducTn-seq identifies C. rodentium* colonization factors” (specifically lines 412-415 in the revised manuscript). We also quantify the phenotype within the host of all the *C. rodentium* terminal oxidases in the new Extended Data Fig. 5 and discuss these complexes in lines 417-425.

Minor comments

1. the text in lines 85-86 is hidden by figure 1.

Strangely, we do not observe this problem in the submitted PDF version of the manuscript. To address this issue, we have included an additional blank line between the text and Fig. 1.

2. Line 320 – EPEC & EHEC – full names of the pathogens should be included.

We have made this correction.

3. Lines 454-456 – Please provide a reference for extra-host lifestyle distinctions. Also, obligate anaerobe bacteria can also be pathogens (*Clostridium* species *difficile*, *tetani*, *botulinum*, *perfringens*, *Bacteroides fragilis*..), and commensal can grow aerobically (*E. coli*, *Klebsiella*..). The sentence should be phrased more accurately.

We thank the Referee for pointing out this flawed distinction between pathogens and commensals. We have replaced the final section of the Results “*The benefits of plasticity in oxygen-related metabolism*” with a new section “*A toxin addicts C. rodentium to its CRISPR system during infection*” in which we characterize the novel role for CRISPR in *C. rodentium* infection biology.

Referee #3 (review in black, response in blue)

Basta et al describes the development of InducTn-seq, which is a method that allows genome-wide transposon insertion mutations in an inducible manner. This system leverages initial genomic integration into an att site, and subsequent Tn5 copy-paste random genome wide insertions of the cassette, controlled with arabinose induction. The authors tested this system first in *E. coli* in cultures to demonstrate the sensitivity of the method to detect insertions into essential genes. The

system was subsequently tested in different proteobacteria to show versatility. In a mouse pathogen colonization model with *C. rodentium*, the authors demonstrated the use of the system to identify gut colonization factors including those associated with oxygen-related metabolism during the course of *C. rodentium* induced enteric infection.

The work tries to address a known challenge in traditional Tn-seq system, which is that environments that bottleneck a Tn-seq population cannot be easily studied. The proposed solution can overcome this issue by using an inducible transposition system. Overall, the manuscript is clearly written and the results are adequately interpreted from well-designed experiments. The innovation of the system is rather straightforward and is likely to be useful for others in the field studying microbial colonization or pathogenesis. A few questions that the authors could clarify are as follows:

We thank the Referee for their positive comments on the manuscript.

1. The kinetics of the transposition in the system. How many transpositions is expected to occur per generation? The authors describe the accumulation of multiple transpositions in Fig 2d, but it is unclear how long it took to accumulate those. Is the expectation that continual passaging of the transposon will continually accumulate even more transposons over time? A better discussion/treatment of this could be helpful.

Thank you for identifying this point for clarification. The data presented in Fig. 2d is derived from transconjugants carrying the Tn5 transposition complex grown overnight on solid LB agar containing 0.2% arabinose. We have updated the text on line 176 of the revised manuscript to include these experimental details.

To address the Referee's questions, we have performed additional experiments to measure the population-level transposition frequency and frequency of multiple transposition events in *C. rodentium* during colonization (Fig. 5b and c, respectively). These new results are described on lines 357-375 of the revised manuscript.

Interestingly, although there are likely many more generations of growth after five days of arabinose induction in mice compared to overnight growth on LB agar, the *in vivo* and *in vitro* results are remarkably similar. Approximately 42% of *C. rodentium* cells underwent at least one transposon event on either day 8 of infection or after one round of overnight growth on LB agar supplemented with arabinose (compare mouse plus arabinose to culture plus arabinose in Fig. 5b). Similarly, the number of transposition events per cell is qualitatively similar between day 8 of *C. rodentium* infection and overnight induction of *E. coli* in culture (5/9 *C. rodentium* colonies with more than one insertion in Fig. 5c compared to 4/10 *E. coli* colonies in Fig. 2d). We are uncertain of the mechanisms that limit runaway transposition within the cell and hope that further mechanistic research into transposon biology may illuminate this finding.

2. In the absence of arabinose, there is still some residual transposition presumably from leaky expression of pBAD. Could the use of more stringent inducible expression system reduce the background? More importantly, how does this leaky mutagenesis affect the initial genomic background of the population in the screen? The animal to animal correlation seem to be high, but some discussion of this would be good.

Thank you for identifying this point for further discussion. We now measure transposition frequency of *C. rodentium* in the inoculum, during culture, and during infection with or without arabinose in the new Fig. 5b. Our results indicate that the animal inoculum contained few mutants (<1% of the population) and that few new mutants were generated when *C. rodentium* was expanded on LB agar in the absence of induction (compare inoculum to culture minus arabinose in Fig. 5b). These results are described on lines 357-368 of the revised manuscript. We have added additional language addressing background transposition throughout the manuscript, and now include the following text to address the Referee's concern on lines 643-645 in the Discussion: "...tight promoter regulation is important to ensure that in the absence of inducer the effects of selection are not obscured by ongoing background mutagenesis".

As discussed in our response to Referee #1, the primary concern regarding leaky expression of the PBAD promoter in mouse experiments is that new mutants arising during outgrowth of the fecal population on LB could confound phenotypic interpretation. However, our new experimental results (Fig. 5b) find that most mutants in the fecal population arise because of arabinose induction within the mouse rather than during subsequent LB expansion. Moreover, new mutants arising during LB outgrowth are not an issue empirically; if background transposition was confounding our results, InducTn-seq would not be capable of identifying colonization-specific factors such as genes within the LEE pathogenicity island (Fig. 5g). LEE genes contribute to *C. rodentium* fitness during colonization but not during culture. Therefore, if significant numbers of new mutants were created in the LEE during outgrowth, we would not be able to identify the LEE genes as colonization factors.

3. In the *C. rodentium* model, how many mutations were accumulated per cell over time? By the 8th day, one presumes that there are multiple insertional mutations per cell yet the fitness measurements is an integration across all of those mutations. There might be some epistatic effects of multiple mutations that are being observed. Some analysis or discussions of this issue may be warranted.

Thank you for identifying this point for further discussion. In response to the Referee's requests we now measure the number of transposition events per cell on day 8 of infection and include these results in the new Fig. 5c. The number of cells with multiple transposon insertions is qualitatively similar to that observed after overnight induction of *E. coli* in culture (5/9 *C. rodentium* colonies with more than one insertion in Fig. 5c compared to 4/10 *E. coli* colonies in Fig. 2d). These new results are described on lines 370-375 of the revised manuscript.

We have also expanded our discussion of why Tn-seq is robust to the confounding effects of multiple mutations within a cell on lines 549-582 of the revised manuscript and include it here:

"Historically, inducible mutagenesis has been avoided in functional genetic screens presumably to prevent the accumulation of multiple insertions within the same cell. Multiple insertions can obscure the causal relationship between each insertion and an observed phenotype and introduce the possibility of phenotypes arising from genetic interactions between insertions. Therefore, having a single, stably integrated transposon is crucial for interpreting the phenotype of an isolated clone. However, pooled genetic screens like Tn-seq are inherently robust against the potential confounding

effects of multiple transposon insertions within the same cell because the relationship between genotype and phenotype is predicated on the collective behavior of tens to hundreds of unique, independently derived insertions, as described below.

First, leveraging multiple unique insertions eliminates the confounding effects of additional insertions in the cell on interpreting causality. Even in an extreme scenario where an entire population consists of double mutants (i.e., two transposons inserted randomly in the genome of every cell), the phenotypic interpretation for a given gene remains unaffected. Any observed change in a gene's insertion frequency is overwhelmingly due to the insertions in the gene itself, rather than the other random insertions, which are diluted across the rest of the genome. Thus, the collective behavior of mutants in the gene will be attributed to the gene.

Moreover, using non-parametric statistical tests in Tn-seq analysis to compare the concordant behavior of many unique insertion sites (e.g., Mann-Whitney U test⁵) limits the ability of outliers, particularly those caused by positive genetic interactions, to confound interpretation. For example, in a hypothetical condition where a genetic interaction leads to a very strong selective advantage, such as a condition where most mutants cannot grow except for the double mutant, the interaction would appear as an unreproducible 'spike' of insertions at one or a few unique sites within each interacting gene. This finding is generally classified as non-significant by non-parametric statistical tests because the insertion frequency at this site is not concordant with the insertion frequency across the rest of the gene. Consequently, inducible mutagenesis can be used in pooled Tn-seq experiments with minimal risk of multiple insertions in the same cell confounding the results. However, if the transposon mutant population is created in a mutant rather than a wild-type background, then the role of both negative (i.e., synthetic sick or lethal) and positive genetic interactions can be assessed.”